# Minimax Posterior Contraction Rates for Unconstrained Distribution Estimation on $[0,1]^d$ under Wasserstein Distance

**Peter Jacobs**  *pmjacob@sandia.gov*
*School of Computing*
*University of Utah*

**Lekha Patel**  *lpatel@sandia.gov*
*Scientific Machine Learning Department*
*Sandia National Laboratories*

**Anirban Bhattacharya**  *anirbanb@stat.tamu.edu*
*Department of Statistics*
*Texas A&M University*

**Debdeep Pati**  *dpati2@wisc.edu*
*Department of Statistics*
*University of Wisconsin*

**Reviewed on OpenReview:** *https://openreview.net/forum?id=UrSgGSTM2J*

## Abstract

We obtain asymptotic minimax optimal posterior contraction rates for estimation of probability distributions on $[0,1]^d$ under the Wasserstein-$p$ metrics using Bayesian Histograms. To the best of our knowledge, our analysis is the first to provide minimax posterior contraction rates for every $p \geq 1$ and problem dimension $d \geq 1$. Our proof technique takes advantage of the conjugacy of the Bayesian Histogram.

## 1 Introduction

The Wasserstein metric is a popular tool for comparing two distributions $\mu$ and $\nu$ defined on a common metric space $(E^d, \| \cdot - \cdot \|_2)$ where $E \subseteq \mathbb{R}$. For $1 \leq p < \infty$, the Wasserstein distance $W_p$ is defined as

$$W_p(\mu_1, \mu_2) := \left( \inf_{\pi \in \mathcal{M}(\mu_1, \mu_2)} \int \|x - y\|_2^p \, d\pi(x,y) \right)^{1/p}, \tag{1}$$

where $\mathcal{M}(\mu_1, \mu_2)$ is the set of couplings of $\mu_1$ and $\mu_2$; specifically the joint probability measures on $E \times E$ with marginals $\mu_1$ and $\mu_2$ respectively. Some benefits of using the Wasserstein metric include its sensitivity to distance in the underlying space, ability to compare distributions regardless of continuity level, and its 1-dimension equivalent representation as the $L^p$ distance between quantile functions, which facilitates quantile function inference (Zhang et al., 2020).

In this paper we study the problem of non–parametrically estimating a distribution $P_0$ on $E^d$ (where $E = [0,1]$) under the Wasserstein metric from $n$ independent and identically distributed (i.i.d) random variables $Y_1, \ldots, Y_n$ drawn from $P_0$. Our focus is on the unconstrained problem; that is, we place no additional assumptions on $P_0$. From the viewpoint of analyzing only frequentist estimators, this is a well studied problem. The frequentist convergence rates of the empirical measure under the expected Wasserstein distance are studied in (Fournier & Guillin, 2015; Singh & Póczos, 2018; Bobkov & Ledoux, 2019; Weed & Bach, 2019) to varying degrees of generality. A consequence of the work of (Singh & Póczos, 2018) is that on the metric space $([0,1]^d, \| \cdot - \cdot \|_2)$, for $d \in \mathbb{N}$, for the class of Borel probability measures, the empirical measure

is minimax optimal (at least up to logarithmic terms) for every $p \geq 1$. Further, the minimax rate is lower bounded by $n^{-1/2p}$ for $d \leq 2p$, and $n^{-1/d}$ for $d > 2p$.

Far less has been done in providing frequentist guarantees for Bayesian statistical procedures when the inferential goal is to estimate a non-parametric distribution underneath a Wasserstein distance. In a non-parametric Bayesian model aimed at inferring a probability distribution on $E^d$, for each sample size $n$, a prior $\Pi_{0n}$ is placed on the space of Borel probability measures on $E^d$. We denote this space $\mathcal{P}_d(E)$. The sample size $n$ posterior distribution, which we denote $\Pi_n(\cdot|Y_1, Y_2, \ldots, Y_n)$, is a regular conditional distribution over $\mathcal{P}_d(E)$ induced from the likelihood and the prior $\Pi_{0n}$. Given a distance function $\tilde{d}$ between probability measures on $E$ (e.g Kullback-Leibler, Hellinger, Wasserstein–1, Total Variation, etc.) we say the sequence of posterior distributions contracts almost surely at the rate $\epsilon_n$ under $P_0$ if $\Pi_n(P \in \mathcal{P}_d(E) : \tilde{d}(P_0, P) \geq \epsilon_n)$ converges almost surely to 0 as $n \to \infty$ when $Y_1, Y_2, \ldots, Y_n \overset{iid}{\sim} P_0$ (Ghosal & Van der Vaart, 2017).

There are other useful but weaker notions of posterior contraction, such as the *in probability* variant which seeks only to show that for every $M_n \to \infty$, $\Pi_n(P \in \mathcal{P}_d(E) : \tilde{d}(P_0, P) \geq M_n \epsilon_n) \to 0$ in probability under $P_0$. We focus on the stronger *almost sure* version in this work because an almost sure PCR of $\epsilon_n$ holding uniformly over every $P_0$ in a finite class $\mathcal{P}_d(E)$ implies the existence of an estimator $\hat{P}_n$ derived from the posterior achieving $\mathbb{E}_{P_0} \tilde{d}(P_0, \hat{P}_n) \lesssim \epsilon_n$ uniformly over $P_0 \in \mathcal{P}_d(E)$. Specifically, an asymptotically minimax, almost sure PCR holding uniformly over the finite class $\mathcal{P}_d(E)$ cannot decay at a faster rate than the minimax rate [1]. It is for this reason that if for every $P_0 \in \mathcal{P}_d(E)$ the *almost sure* Posterior Contraction Rate (PCR) $\epsilon_n$ is achieved, and $\epsilon_n$ is the frequentist minimax rate over $\mathcal{P}_d(E)$, we say the Bayesian method is agnostic to the prior choice in the presence of an infinite amount of data under class $\mathcal{P}_d(E)$. In particular, the practitioner can specify prior knowledge that may be beneficial for inference at small sample sizes when that prior knowledge is correct, but if this prior knowledge is inaccurate, the method will still be competitive (from the minimax perspective) with the best possible estimator when the sample size is sufficiently large. In our work $\mathcal{P}_d(E)$ is not finite, and we introduce in Definition 1 a slightly stronger property of contraction rate across a distribution class which cannot outperform the minimax rate regardless of the size of $\mathcal{P}_d(E)$; the main result of this work proves this stronger property.

Ghosal et al. (2000) provides a general three condition verification strategy for proving these PCRs. This strategy or minor variants have been the catalyst for a myriad of minimax PCR results for the problem of estimating a distribution via i.i.d samples. For example, Scricciolo (2007) uses a histogram model with a prior on the bin weights and number of bins to prove minimax PCRs for the distribution class possessing $\alpha$ Holder smooth densities on $[0,1]^d$, where $\alpha \in (0,1]$, $d \geq 1$ and $\tilde{d}$ is the Hellinger distance. Kruijer & Van der Vaart (2008) also achieves minimax PCRs for the estimation of $\alpha$–Holder density class when $d \in \{1, 2\}$ and $\tilde{d}$ is the Hellinger distance using a weighted mixture of Beta density functions with a prior on the mixture weights and number of mixed densities. A notable achievement of these works is that the value $\alpha$ is not used in the construction of the posterior, yet minimax optimality is still achieved for the $\alpha$ Holder density class. That is, the methods adapt to the smoothness parameter. Shen et al. (2013) provides minimax PCRs for the estimation of smooth distributions on $\mathbb{R}^d$ using a weighted mixture of Gaussian distributions with a prior on the covariance matrix of the mixed Gaussians when $\tilde{d}$ is the Hellinger distance or Total Variation distance; smoothness adaptive guarantees are provided. However, note that rather than studying the unconstrained distribution estimation setting, these works focus on estimation of distributions known to possess a smoothness property, and in any case do not study estimation underneath Wasserstein distance.

Unfortunately the Ghosal et al. (2000) approach is more difficult to use when $\tilde{d}$ is a Wasserstein metric. Challenges include $W_p, p \geq 2$ not being dominated by Total Variation or Hellinger distances, causing the need for explicit test construction. Also, the Kullback-Leibler neighborhood condition, which ensures such neighborhoods of $P_0$ have sufficient prior mass, may make it more difficult to achieve the minimax rate under $W_p, p \geq 1$ because depending on the model under consideration, approximation of distributions under the Kullback-Leibler divergence may not be achievable at the square of minimax rate under $W_p$ [2]. In light of these challenges, there have been far fewer theoretical advances in proving minimax optimal PCRs for distribution estimation under $W_p, p \geq 1$ than under Total Variation and Hellinger distances.

---

[1] see appendix Lemma 9 for proof

[2] Chae et al. (2021) (p.3644) already encounters Kullback-Liebler condition limitations when only estimating distributions on $\mathbb{R}$

Chae et al. (2021) successfully derive a 2 condition verification strategy for proving PCRs under Wasserstein distance when $E = \mathbb{R}$. This work appears to be the first providing a set of general conditions for proving Wasserstein PCRs for the unconstrained, non-parametric distribution estimation problem, and their method is applicable for distributions with unbounded domain. Their results are restricted to the study of 1 dimensional distributions. Also, the Ghosal et al. (2000) and Chae et al. (2021) frameworks depend on the posterior distribution being available through Bayes formula. Camerlenghi et al. (2022) observes that this can be limiting and develops a method to derive PCRs when the posterior is not available via Bayes formula. They apply their technique to derive Wasserstein PCRs for each $d \in \mathbb{N}, v \geq 1$ for the model placing a Dirichlet process prior on the data generating distribution. The PCR derived for $P_0 \in \mathcal{P}_d([0,1])$ is $\gtrsim n^{-\frac{1}{2}\frac{1}{(d+p)}}$ which via the discussion earlier in this section is slower decaying than the minimax rate by a polynomial factor for every $d \in \mathbb{N}, p \geq 1$. In the Deconvolution problem, where for $i \in \{1, 2, \ldots, n\}$, $Y_i = X_i + \epsilon_i$ and $X_1, \ldots, X_n \overset{iid}{\sim} P_0$ is independent of $\epsilon_1, \ldots, \epsilon_n \overset{iid}{\sim} \mu_0$ and $\mu_0$ is known and the goal is to infer $P_0$, Wasserstein distance PCRs have been derived in Rousseau & Scricciolo (2023),Gao & van der Vaart (2016), and Scricciolo (2018). $P_0$ is called the mixing distribution. The distribution of $Y_1$, however, is the convolution $P_0 * \mu_0$.

## 1.1 Contributions

Our main contribution is Theorem 1. In it we obtain PCRs for every dimension $d \geq 1$ and for every distance $W_p, p \geq 1$ and the PCRs achieved are minimax optimal at least up to logarithmic terms. To the best of our knowledge, our result is the first to provide a minimax optimal PCR across each ($d \in \mathbb{N}, p \geq 1$) setting for estimating an unconstrained $P_0 \in \mathcal{P}_d([0,1])$. These rates are achieved using a Bayesian Histogram model that partitions $[0,1]^d$ into $b_n^d$ equal area squares where $b_n := 2^{\lceil \log_2(k_n) \rceil}$ for a sequence $k_n$ growing as a function of the sample size $n$ at the appropriate rate, uses the Multinomial likelihood to weight the constant density within each square, and places a sample size dependent Dirichlet prior distribution on the weight vector with prior concentration vector $\boldsymbol{\alpha}_{b_n}$ (of dimension $b_n^d$). This model induces a sequence of posterior distributions $\Pi_{n,k_n,\boldsymbol{\alpha}_{b_n}}$ over $\mathcal{P}_d([0,1])$. In Theorem 1, we show that

$$\Pi_{n,k_n,\boldsymbol{\alpha}_{b_n}}(P \in \mathcal{P}_d([0,1]) : W_p(P_0, P) \geq \epsilon_n(d,v)) \to 0$$

in a very strong sense under $P_0$ (implying *almost sure* and *in probability* convergence) provided that

1. If $d \leq 2p$ then $k_n = n^{\frac{1}{2v}}$, $\sum_{\boldsymbol{j} \in 2^{d\lceil \log_2(k_n) \rceil}} \boldsymbol{\alpha}_{\boldsymbol{j},b_n} \lesssim n^{\frac{1}{2}}$, $\epsilon_n \asymp n^{-\frac{1}{2p}}$

2. If $d > 2p$, then $k_n = n^{\frac{1}{d}}$, $\sum_{\boldsymbol{j} \in 2^{d\lceil \log_2(k_n) \rceil}} \boldsymbol{\alpha}_{\boldsymbol{j},b_n} \lesssim n^{1-\frac{p}{d}}$, $\epsilon_n \asymp n^{-\frac{1}{d}}$

where log terms in $\epsilon_n$ which are specified in the theorem are ignored here. In the problem of providing optimal PCRs for estimating distributions on $[0,1]^d$ under Wasserstein-$p$ distance, our results close the gap between the minimax rates for this problem and the Wasserstein PCRs provided by Camerlenghi et al. (2022).

The remainder of this paper is organized as follows. In Section 2 we formally introduce the Bayesian Histogram model. In Section 3 we introduce the strong notion of posterior contraction under which our theorems are proved and show connections between this notion of posterior contraction with minimax lower bound rates and traditional PCR statements. In Section 4 we state the main theorem and the three fundamental lemmas upon which the main theorem depends. We then prove the main theorem. In Section 5 we provide instruction on how to use the prior distribution to express prior beliefs when using this model in practice. In Section 6 we provide the proofs of the lemmas and in Section 7 we provide concluding remarks.

## 2 Bayesian Histogram

### 2.1 General Notations

Since we always consider probability distributions on $[0,1)^d$, we drop the $E$ notation of the introduction and denote

$$\mathcal{P}_d := \{\text{Borel Probability Measures on } [0,1)^d\}$$

Excluding the right end points are a notational convenience but extension of the arguments that follow to include the right endpoint is trivial.

For sequences of numbers $a_n, b_n$ defined for sufficiently large $n \in \mathbb{N}$, $a_n \lesssim b_n$ means there exists a $C > 0$ and $N_0 \in \mathbb{N}$ such that for $n \geq N_0, a_n \leq Cb_n$. For $b, d \in \mathbb{N}$, we denote $[b] := \{1, 2, \ldots, b\}$ and $[b]^d := \prod_{j=1}^{d}[b]$. For $B \subseteq \mathbb{R}^d$, $\mathcal{B}(B)$ denotes the Borel measurable subsets of $B$. For $j \in \mathbb{N}$, $\mathcal{S}^{j-1}$ refers to the $(j-1)$ dimensional probability simplex. That is $\mathcal{S}^{j-1} := \{(x_1, \ldots, x_j) \in \mathbb{R}^j : \sum_{t=1}^{j} x_t = 1, x_t \geq 0 \text{ for } t \in [j]\}$. Also note that $\mathbb{R}_+ := \{x \in \mathbb{R} : x > 0\}$ and for $z \in \mathbb{N}$ and $\boldsymbol{\alpha} \in \mathbb{R}_+^z$, the Dirichlet probability measure Dirichlet $: \mathcal{B}(\mathcal{S}^{z-1}) \to [0, 1]$ is given by

$$\text{Dirichlet}(G|\boldsymbol{\alpha}) = \frac{1}{B(\boldsymbol{\alpha})} \int_G \prod_{i=1}^{z} x_i^{\alpha_i - 1} \mathrm{d}\boldsymbol{x}, \tag{2}$$

where $B(\boldsymbol{\alpha} = (\alpha_1, \alpha_2, \ldots, \alpha_z)) := \frac{\prod_{j=1}^{z} \Gamma(\alpha_j)}{\Gamma(\sum_{j=1}^{z} \alpha_j)}$ is the $z$ dimensional Beta function and $\Gamma(x)$ denotes the Gamma function evaluated at $x$ and $G \in \mathcal{B}(\mathcal{S}^{z-1})$. For $b \in \mathbb{N}$ and a multi-index $\boldsymbol{i} = (i_1, i_2, \ldots, i_d) \in [b]^d$, define

$$A_{\boldsymbol{i},b} := \left[\frac{i_1 - 1}{b}, \frac{i_1}{b}\right) \times \left[\frac{i_2 - 1}{b}, \frac{i_2}{b}\right) \times \ldots \times \left[\frac{i_d - 1}{b}, \frac{i_d}{b}\right). \tag{3}$$

Clearly, $\{A_{\boldsymbol{i},b}\}_{\boldsymbol{i} \in [b]^d}$ form a partition of $[0, 1)^d$. For a vector of weights $\boldsymbol{\pi} = \{\pi_{\boldsymbol{j}}\}_{\boldsymbol{j} \in [b]^d} \in \mathcal{S}^{bd-1}$, the $d$ dimensional Histogram probability measure Histogram $: \mathcal{B}([0, 1)^d) \to [0, 1]$ is a weighted mixture of uniform distributions on the partition sets $A_{\boldsymbol{i},b}$, defined by

$$\text{Histogram}(G|\boldsymbol{\pi}, b) := \int_G \sum_{\boldsymbol{i} \in [b]^d} b^d \pi_{\boldsymbol{i}} \mathbb{I}(\boldsymbol{y} \in A_{\boldsymbol{i},b}) \mathrm{d}\boldsymbol{y}, \tag{4}$$

where $G \in \mathcal{B}([0, 1)^d)$.

## 2.2 Bayesian Histogram Model Definition

We suppose $Y_1, Y_2, \ldots, Y_n \overset{iid}{\sim} P_0$ where $P_0 \in \mathcal{P}_d$. For $b \in \mathbb{N}$, let $\boldsymbol{\alpha}_b := \{\alpha_{\boldsymbol{j},b}\}_{\boldsymbol{j} \in [b]^d} \in \mathbb{R}_+^{bd}$. For an increasing sequence $k_n$, let $b_n := 2^{K_n}$, where $K_n := \lceil \log_2(k_n) \rceil$, $\boldsymbol{\pi}_n := \{\pi_{n,\boldsymbol{j}}\}_{\boldsymbol{j} \in [b_n]^d} \in \mathcal{S}^{b_n d - 1}$. For $n \in \mathbb{N}$, the Bayesian Histogram model likelihood and prior are given by

$$Y_1, \ldots, Y_n | \boldsymbol{\pi}_n \overset{i.i.d}{\sim} \text{Histogram}(\cdot | \boldsymbol{\pi}_n, b_n), \qquad \boldsymbol{\pi}_n | \boldsymbol{\alpha}_{b_n} \sim \text{Dirichlet}(\cdot | \boldsymbol{\alpha}_{b_n}). \tag{5}$$

Also, let $z_n^*(\cdot|Y_1, \ldots, Y_n)$ refer to the posterior probability measure over $\mathcal{S}^{b_n d - 1}$ derived from Equation 5. As $\alpha_{\boldsymbol{i},b_n} > 0$ for every $\boldsymbol{i} \in [b_n]^d$ and for every $n \in \mathbb{N}$, Equation 5 induces a sequence of posterior distributions over $\mathcal{P}_d$. Specifically let $\psi_b : \mathcal{S}^{bd-1} \to \mathcal{P}_d$ be the map that takes a given $\boldsymbol{\pi} = \{\pi_{\boldsymbol{j}}\}_{\boldsymbol{j} \in [b]^d}$ and produces its corresponding Histogram probability measure. That is

$$\psi_b(\boldsymbol{\pi}) = \text{Histogram}(\cdot | \boldsymbol{\pi}, b). \tag{6}$$

For a measurable set $B \subseteq \mathcal{P}_d$, the posterior measure $\Pi_{n, k_n, \boldsymbol{\alpha}_{b_n}}$ is

$$\Pi_{n, k_n, \boldsymbol{\alpha}_{b_n}}(B|Y_1, \ldots, Y_n) = z_n^*(\psi_{b_n}^{-1}(B)|Y_1, \ldots, Y_n). \tag{7}$$

Because the conjugate prior of the Multinomial distribution is the Dirichlet distribution (see for example Gelman et al. (2013), Section 3.4) and the induced likelihood over the vector of bin counts is the Multinomial distribution, it is straightforward to show that

$$z_n^*(\cdot|Y_1, \ldots, Y_n) = \text{Dirichlet}(\cdot | \boldsymbol{\alpha}_{\boldsymbol{b_n}}^*),$$

where for $\boldsymbol{i} \in [b_n]^d$

$$\alpha_{\boldsymbol{i},b_n}^* = \alpha_{\boldsymbol{i},b_n} + \sum_{j=1}^{n} \mathbb{I}(Y_j \in A_{\boldsymbol{i},b_n}). \tag{8}$$

Now allowing $\boldsymbol{\alpha}_{b_n} \in \{x \in \mathbb{R} : x \geq 0\}^{b_n d}$, we define the sequence of estimators for $P_0$, denoted $\bar{P}_n$, by

$$\bar{P}_{n,k_n,\boldsymbol{\alpha}_{b_n}} := \psi_{b_n} \left\{ \left( \frac{\alpha^*_{\boldsymbol{i},b_n}}{\sum_{\boldsymbol{j} \in [b_n]^d} \alpha^*_{\boldsymbol{j},b_n}} \right)_{\boldsymbol{i} \in [b_n]^d} \right\} = \psi_{b_n} \{ (E_{z_n^*}(\pi_{\boldsymbol{i}} | Y_1, \ldots, Y_n))_{\boldsymbol{i} \in [b_n]^d} \}, \tag{9}$$

where the second equality above holds if $\alpha_{\boldsymbol{i},b_n} > 0$ for $\boldsymbol{i} \in [b_n]^d$.

We note that posterior distributions derived from improper prior distributions are not considered in this work, and therefore to consider the posterior measure sequence $\Pi_n$ we require that $\alpha_{\boldsymbol{i},b_n} > 0$ for $\boldsymbol{i} \in [b_n]^d$. However, we allow $\bar{P}_n$ to be defined regardless of whether or not the prior distribution over the simplex is proper. In particular, it is still defined in the event that some or all of the $\alpha_{\boldsymbol{i},b_n}$ parameters are zero. When the prior distribution is proper, $\bar{P}_n$ has an additional interpretation: it is the Posterior Mean Histogram. In the lemmas and theorems that follow that involve analysis of the posterior distribution sequence $\Pi_n$, we make clear that we require $\alpha_{\boldsymbol{i},b_n} > 0$ for $\boldsymbol{i} \in [b_n]^d$ and $n \in \mathbb{N}$.

$\bar{P}_{n,k_n,\boldsymbol{\alpha}_{b_n}}$ (and $\Pi_{n,k_n,\boldsymbol{\alpha}_{b_n}}$) are indexed by the choice of $k_n$ (which determines the total number of bins) and $\boldsymbol{\alpha}_{b_n}$, which gives the prior concentrations on those bins. In the subsequent subsection we establish constraints on $k_n$ and $\boldsymbol{\alpha}_{b_n}$ that ensure $\bar{P}_{n,k_n,\boldsymbol{\alpha}_{b_n}}$ and $\Pi_{n,k_n,\boldsymbol{\alpha}_{b_n}}$ are minimax statistical procedures.

## 3   Notions of Minimax Posterior Contraction

First we introduce a strong notion of posterior contraction across an entire distribution class which cannot decay faster than the minimax rate in general (regardless of the size of the space). It is for this reason we call this stronger notion of posterior contraction a *minimax-conscious* PCR.

**Definition 1.** *(minimax-conscious PCR) Let $\tilde{d}$ be a distance over $\mathcal{P}_d$ and $\Pi_n$ some sequence of posterior distributions over $\mathcal{P}_d$. The sequence $\epsilon_n$ is called a minimax-conscious PCR for the sequence of posterior distributions $\Pi_n$ on the space $(\mathcal{P}_d, \tilde{d})$ if there exists sequence $z_n$ such that $z_n \to 0$ and sequence $\delta_n$ such that $\delta_n \lesssim \epsilon_n, \sum_{j=1}^{\infty} \delta_n < \infty$ and for some $N$ sufficiently large, and $n \geq N$, for every $P_0 \in \mathcal{P}_d$, whenever $Y_1, \ldots, Y_n \overset{iid}{\sim} P_0$,*

$$P_0 \left( \Pi_n(P \in \mathcal{P}_d : \tilde{d}(P, P_0) \geq \epsilon_n) \leq z_n \right) \geq 1 - \delta_n.$$

Now we make the connection between *minimax-conscious PCRs* and minimax lower bounds.

**Lemma 1.** *If $\epsilon_n$ is a minimax-conscious PCR for a sequence of posterior distributions $\Pi_n$ on the space $(\mathcal{P}_d, \tilde{d})$ and distance $\tilde{d}$ is bounded above on $\mathcal{P}_d$ and $m_n$ satisfies*

$$\inf_{\tilde{P}} \sup_{P_0 \in \mathcal{P}_d} \mathbb{E}_{P_0} \tilde{d}(P_0, \tilde{P}) \gtrsim m_n,$$

*where the inf is taken over all function of $n$ iid samples, then $\epsilon_n \gtrsim m_n$. In particular, a minimax-conscious PCR can never decay faster than the minimax rate.*

Note that a minimax-conscious PCR is a statement about the behavior of a posterior distribution sequence across an entire distribution class. Next we make the connection between a minimax-conscious PCR and the traditional, *almost sure* PCR definition discussed in the introduction. Specifically, a minimax-conscious PCR implies an *almost sure* PCR at the same rate for every distribution in the class.

**Lemma 2.** *If $\epsilon_n$ is a minimax-conscious PCR for the sequence of posterior distributions $\Pi_n$ on the space $(\mathcal{P}_d, \tilde{d})$, then for every $P_0 \in \mathcal{P}_d$, whenever $Y_1, \ldots, Y_n \overset{iid}{\sim} P_0$,*

$$\Pi_n(P \in \mathcal{P}_d : \tilde{d}(P, P_0) \geq \epsilon_n) \to 0 \; P_0 \; almost \; surely \; as \; n \to \infty.$$

The proofs of Lemmas 1 and 2 are contained in Appendix 8.1. Note that Lemma 1 only requires that $z_n < \frac{1}{2}$ eventually and does not use that $\sum_{j=1}^{\infty} \delta_n < \infty$. The criteria that $z_n \to 0$ and $\sum_{j=1}^{\infty} \delta_n < \infty$ are only used in Lemma 2.

## 4 Posterior Contraction Results

Our results utilize the following two assumptions for $d \in \mathbb{N}$ and $p \geq 1$.

**Assumption 1.** *For $n \in \mathbb{N}$*

$$k_n = \begin{cases} n^{1/2p} & d \leq 2p, \\ n^{1/d} & d > 2p, \end{cases}$$

and

**Assumption 2.**

$$\sum_{\boldsymbol{j} \in [b_n]^d} \alpha_{\boldsymbol{j}, b_n} \lesssim \begin{cases} n^{1/2} & d \leq 2p. \\ n^{1 - \frac{v}{d}} & d > 2p. \end{cases}$$

Our main PCR result is the following theorem.

**Theorem 1.** *Suppose $\gamma > 1$ and $k_n$ satisfies Assumption 1 and*

$$\epsilon_n(d, p) := C_0(d, p) \begin{cases} n^{-\frac{1}{2p}} \log^{\frac{\gamma}{p}}(n) & d < 2p, \\ n^{-\frac{1}{2p}} \log^{\frac{1+\gamma}{p}}(n) & d = 2p, \\ n^{-\frac{1}{d}} \log^{\frac{\gamma}{p}}(n) & d > 2p, \end{cases} \tag{10}$$

*Now assuming that for each $n \in \mathbb{N}$ and $\boldsymbol{j} \in [b_n]^d$, $\alpha_{\boldsymbol{j}, b_n} > 0$, and that $\boldsymbol{\alpha}_{b_n}$ satisfies Assumption 2, we have that for $1 \leq p < \infty$ and $d \in \mathbb{N}$ and $C_0(d, p)$ sufficiently large, $\epsilon_n(d, p)$ is a minimax-conscious PCR for the sequence of posterior distribution $\Pi_{n, k_n, \boldsymbol{\alpha}_{b_n}}$ on the space $(\mathcal{P}_d, W_p)$ where $b_n = 2^{\lceil \log_2(k_n) \rceil}$.*

According to Singh & Póczos (2018),

$$\inf_{\tilde{P}} \sup_{P_0 \in \mathcal{P}_d} \mathbb{E}_{P_0} W_v(\tilde{P}, P_0) \gtrsim \begin{cases} n^{-\frac{1}{2p}} & d \leq 2p, \\ n^{-\frac{1}{d}} & d > 2p, \end{cases} \tag{11}$$

where the inf is taken over all estimators $\tilde{P}$ from $n$ observations. Thus the PCRs of Theorem 1 are up to logarithmic terms attaining the minimax rates. The assumption on the prior concentrations, Assumption 2, is flexible enough to support a vague prior. Specifically, the mean of a Dirichlet distribution with common concentration on all categories is a discrete uniform distribution, so the practitioner wishing to encode vagueness by asserting that under the prior on average all bin probabilities are equal will want to set all prior bin concentrations to a common value. When $d \leq 2p$, by Assumption 1, the number of bins is $\asymp n^{\frac{d}{2p}}$, thus Assumption 2 is satisfied when each concentration is set to $Cn^{-(\frac{d}{2p} - \frac{1}{2})}$ for some $C > 0$. Likewise when $d > 2p$, by Assumption 1, there are $\asymp n$ bins and Assumption 2 is satisfied when all concentrations are $Cn^{-\frac{p}{d}}$. Also note that while Assumption 2 places an upper bound on the total volume of the prior concentrations to ensure the prior does not overwhelm the empirical Histogram at large sample sizes, it in general does not place any shape restrictions on the prior; in particular other prior shapes besides the uniform can be constructed.

The proof of Theorem 1 is composed from the following three auxiliary lemmas. The first auxiliary lemma upper bounds the rate of convergence of the posterior mean histogram, $\bar{P}_{n, k_n, \boldsymbol{\alpha}_{b_n}}$, towards $P_0$ in mean $W_p^p$ distance. The second lemma establishes an exponentially decaying upper bound on the probability the $W_p^p$ distance between the posterior mean histogram and $P_0$ deviates from its mean by more than $\epsilon > 0$. The third lemma establishes a PCR *around* $\bar{P}_{n, k_n, \boldsymbol{\alpha}_{b_n}}$, rather than $P_0$. It is the third lemma that leverages the conjugacy of this model.

**Lemma 3.** *Let $Y_1, \ldots, Y_n \overset{iid}{\sim} P_0 \in \mathcal{P}_d$. Suppose $k_n$ satisfies Assumption 1, $\boldsymbol{\alpha}_{b_n}$ satisfies Assumption 2 and that for $n \in \mathbb{N}$ and $\boldsymbol{j} \in [b_n]^d$, $\alpha_{\boldsymbol{j}, b_n} \geq 0$. Then*

$$\sup_{P_0 \in \mathcal{P}_d} \mathbb{E}_{P_0} W_p^p(P_0, \bar{P}_{n, k_n, \boldsymbol{\alpha}_{b_n}}) \lesssim \begin{cases} n^{-\frac{1}{2}} & d < 2p. \\ n^{-\frac{1}{2}} \log(n) & d = 2p. \\ n^{-\frac{p}{d}} & d > 2p. \end{cases}$$

**Lemma 4.** *Let $Y_1, \ldots, Y_n \overset{iid}{\sim} P_0 \in \mathcal{P}_d$. Suppose $k_n$ satisfies Assumption 1 and $\alpha_{\boldsymbol{j}, b_n} > 0$ for each $n \in \mathbb{N}$ and $\boldsymbol{j} \in [b_n]^d$. Then for $1 \leq p < \infty$ and $d \in \mathbb{N}$ and $\epsilon > 0$*

$$\sup_{P_0 \in \mathcal{P}_d} P_0 \left( W_p^p \left( P_0, \bar{P}_{n,k_n,\boldsymbol{\alpha}_{b_n}} \right) - \mathbb{E}_{P_0} W_p^p(P_0, \bar{P}_{n,k_n,\boldsymbol{\alpha}_{b_n}}) \geq \epsilon \right) \leq \exp(-2d^{-p} n \epsilon^2)$$

**Lemma 5.** *Let $Y_1, \ldots, Y_n \overset{iid}{\sim} P_0 \in \mathcal{P}_d$. Suppose $k_n$ satisfies Assumption 1. Let $\gamma > 1$, and let $\{\tau_n(d,p)\}_{n=1}^{\infty}$ be a sequence satisfying*

$$\tau_n(d,p) = C_1(d,p) \begin{cases} n^{-\frac{1}{2p}} \log^{\frac{\gamma}{p}}(n) & d < 2p, \\ n^{-\frac{1}{2p}} \log^{\frac{1+\gamma}{p}}(n) & d = 2p, \\ n^{-\frac{1}{d}} \log^{\frac{\gamma}{p}}(n) & d > 2p, \end{cases} \tag{12}$$

*Then, provided that $\alpha_{\boldsymbol{j}, b_n} > 0$ for each $n \in \mathbb{N}$ and $\boldsymbol{j} \in [b_n]^d$, we have that for $1 \leq p < \infty$ and $d \in \mathbb{N}$, there exists $C_1(d,p), N_1(d,p)$ sufficiently large such that for each $P_0 \in \mathcal{P}_d$,*

$$\Pi_{n,k_n,\boldsymbol{\alpha}_{b_n}} (P \in \mathcal{P}_d : W_v(P, \bar{P}_{n,k_n,\boldsymbol{\alpha}_{b_n}}) \geq \tau_n(d,p)) \leq 2 \log^{1-\gamma}(n)$$

*almost surely under $P_0$ whenever $n \geq N_1(d,v)$.*

The main technical challenges appear in proving the auxiliary lemmas. Given Lemmas 3, 4 and 5, Theorem 1 follows easily and we show this now. For ease in notation, through the remainder of the paper we drop the $k_n$ and $\boldsymbol{\alpha}_{b_n}$ subscripts from the notation for the posterior, thus $\Pi_{n,k_n,\boldsymbol{\alpha}_{b_n}}$ is referred to as $\Pi_n$ (and $\bar{P}_{n,k_n,\boldsymbol{\alpha}_{b_n}}$ is referred to as $\bar{P}_n$). This does not cause ambiguity in what follows because the values of $k_n$ and $\boldsymbol{\alpha}_{b_n}$ are given in Assumptions 1 and 2.

*Proof of Theorem 1.* By the triangle inequality and union bound, for each $P_0 \in \mathcal{P}_d$,

$$\Pi_n (P \in \mathcal{P}_d : W_p(P_0, P) \geq \epsilon_n(d,p)) \leq \Pi_n \left( P \in \mathcal{P}_d : W_p(P_0, \bar{P}_n) \geq \frac{\epsilon_n(d,p)}{2} \right)$$

$$+ \Pi_n \left( P \in \mathcal{P}_d : W_p(P, \bar{P}_n) \geq \frac{\epsilon_n(d,p)}{2} \right)$$

$$= \mathbb{I} \left[ W_p(P_0, \bar{P}_n) \geq \frac{\epsilon_n(d,p)}{2} \right]$$

$$+ \Pi_n \left( P \in \mathcal{P}_d : W_p(P, \bar{P}_n) \geq \frac{\epsilon_n(d,p)}{2} \right). \tag{13}$$

To handle the first term on the right hand side of Equation 13, first note that by definition of $\epsilon_n(d,p)$ (see Equation 10) and Lemma 3, for each $d \in \mathbb{N}, 1 \leq p < \infty$, we have that $\sup_{P_0 \in \mathcal{P}_d} \mathbb{E}_{P_0} W_p^p(P_0, \bar{P}_n) = o\left(\epsilon_n^p(d,p)\right)$. In particular, there is an $N$ such that for $n \geq N$, and every $P_0 \in \mathcal{P}_d$, whenever $Y_1, \ldots, Y_n \overset{iid}{\sim} P_0$,

$$P_0 \left( W_p^p(P_0, \bar{P}_n) \geq \frac{\epsilon_n^p(d,p)}{2^p} \right) = P_0 \left( W_p^p(P_0, \bar{P}_n) - \mathbb{E}_{P_0} W_p^p(P_0, \bar{P}_n) \geq \frac{\epsilon_n^p(d,p)}{2^p} - \mathbb{E}_{P_0} W_p^p(P_0, \bar{P}_n) \right)$$

$$\leq P_0 \left( W_p^p(P_0, \bar{P}_n) - \mathbb{E}_{P_0} W_p^p(P_0, \bar{P}_n) \geq \frac{\epsilon_n^p(d,p)}{2^{p+1}} \right)$$

$$\leq \exp \left( -2^{-(2p+1)} d^{-p} n \epsilon_n^{2p}(d,p) \right), \tag{14}$$

where in the last inequality above we have applied Lemma 4 with $\epsilon = \frac{\epsilon_n^p(d,p)}{2^{p+1}}$. Now again using the definition of $\epsilon_n^p(d,p)$ and also that $\gamma > 1$, we have that

$$\exp \left( -2^{-(2p+1)} d^{-p} n \epsilon_n^{2p}(d,p) \right) \lesssim \exp \left( -2^{-(2p+1)} d^{-p} C_0^{2p}(d,p) \log(n) \right) \leq \frac{1}{n^2}, \tag{15}$$

where the last inequality is provided $C_0(d,p) \geq 2^{1+\frac{1}{p}} d^{1/2}$ and thus $2^{-(2p+1)} d^{-p} C_0^{2p}(d,p) \geq 2$. By Equations 14 and 15 we have that

$$\sup_{P_0 \in \mathcal{P}_d} P_0 \left( W_p(P_0, \bar{P}_n) \geq \frac{\epsilon_n(d,p)}{2} \right) \lesssim n^{-2} \tag{16}$$

for $d \in \mathbb{N}, p \geq 1$ and $C_0(d, p) \geq 2^{1 + \frac{1}{p}} d^{1/2}$. To handle the second term in the right hand side of Equation 13, note that setting $C_0(d, p) \geq 2C_1(d, p)$ we have that $\tau_n(d, p) \leq \frac{\epsilon_n(d,p)}{2}$ for every $p \geq 1, d \in \mathbb{N}$ where $\tau_n(d, p)$ is as defined in Lemma 5. Using this and Lemma 5, we have that for every $p \geq 1, d \in \mathbb{N}$, an $N(d, p)$ such that for each $P_0 \in \mathcal{P}_d$, when $Y_1, \ldots, Y_n \overset{iid}{\sim} P_0$,

$$\Pi_n \left( P \in \mathcal{P}_d : W_p(P, \bar{P}_n) \geq \frac{\epsilon_n(d, p)}{2} \right) \leq \Pi_n \left( P \in \mathcal{P}_d : W_p(P, \bar{P}_n) \geq \tau_n(d, p) \right) \leq 2 \log^{1-\gamma}(n) \tag{17}$$

with probability 1 under $P_0$ when $n \geq N(d, p)$. By Equations 13, 16, and 17, we conclude the existence of a sequence $\delta_n \asymp n^{-2}$ and a single $N(d, p)$ such that for $n \geq N(d, p)$ and each $P_0 \in \mathcal{P}_d$, whenever $Y_1, \ldots, Y_n \overset{iid}{\sim} P_0$,

$$P_0 \left( \Pi_n \left( P \in \mathcal{P}_d : W_p(P_0, P) \geq \epsilon_n(d, p) \right) \leq 2 \log^{1-\gamma}(n) \right) \geq 1 - \delta_n \tag{18}$$

for $d \in \mathbb{N}, p \geq 1$ as long as $C_0(d, p) \geq \max(2C_1(d, p), 2^{1 + \frac{1}{p}} d^{1/2})$. Finally note that $2 \log^{1-\gamma}(n) \to 0$ since $\gamma > 1$ and also $\sum_{j=1}^{\infty} \delta_n < \infty$ and $\delta_n = o(\epsilon_n(d, p))$ since $\delta_n \asymp n^{-2}$. The theorem statement thus follows. $\square$

In practice, the Posterior Mean Histogram, introduced in Equation 9 and shown to achieve the same rates for estimation as the posterior itself in Lemma 3 (after application of Jensen's inequality), can be used as the functional point estimator for the unknown distribution.

## 5  Using Prior Information

As discussed in the introduction, minimax optimal posterior contraction results are an indicator that a Bayesian method is (in the worst case risk sense) robust to the selection of an inaccurate prior when the sample size is large.

The general benefit of the prior is in scenarios where the practitioner has a belief about $P_0$ before data collection, is able to encode this information through the prior, and the belief ends up being correct, in which case the small sample size performance can be better than under a purely frequentist estimation approach.

The type of belief that is representable through the prior in this model is a hypothesis about the probability distribution of a partition of $[0, 1]^d$. Specifically, suppose the practitioner believes, but is not certain, that $P_0$ satisfies that on a size $M$ partition of $[0, 1]^d$, denoted $\{R_j\}_{j=1}^M$, the probability in region $R_j$ is $p_j$ for $j \in [m]$. Further suppose there exists a $k_0 \in \{1, 2, \ldots\}$ sufficiently large such that there exists a size $M$ partition of $\{\boldsymbol{i}\}_{\boldsymbol{i} \in [2^{k_0}]^d}$, denoted $\{I_{j,k_0}\}_{j=1}^M$ and for each $j \in [M]$

$$R_j = \bigcup_{\boldsymbol{i} \in I_{j,k_0}} A_{\boldsymbol{i}, 2^{k_0}},$$

where recall $A_{\boldsymbol{i}, 2^{k_0}}$ is defined in Equation 3. In words, each region $R_j$ is expressible as unions of members of the level $k_0$ dyadic partition. Note that because $R_j$ is expressible as a union of members of the partition $\{A_{\boldsymbol{i}, 2^{k_0}}\}_{\boldsymbol{i} \in [2^{k_0}]^d}$ and the partitions $\{A_{\boldsymbol{i}, 2^k}\}_{\boldsymbol{i} \in [2^k]^d}$ for $k \geq k_0$ are nested in the $\{A_{\boldsymbol{i}, 2^{k_0}}\}_{\boldsymbol{i} \in [2^{k_0}]^d}$ partition, $R_j$ is also expressible as a union of members of $\{A_{\boldsymbol{i}, 2^k}\}_{\boldsymbol{i} \in [2^k]^d}$ for $k \geq k_0$. In particular for $k \geq k_0$, there is a size $M$ partition of $\{\boldsymbol{i}\}_{\boldsymbol{i} \in [2^{k_0}]^d}$, denoted $\{I_{j,k}\}_{j=1}^M$ and for each $j \in [M]$, $R_j = \bigcup_{\boldsymbol{i} \in I_{j,k}} A_{\boldsymbol{i}, 2^k}$. We recommend the following procedure for incorporating prior knowledge in these circumstances:

1. When $k_n$ of Assumption 1 satisfies $\lceil \log_2(k_n) \rceil < k_0$, reset $k_n := k_0$. Otherwise use $k_n$ as in Assumption 1. This small sample size analysis decision has no impact on the large sample (asymptotic) results of Theorem 1.

2. For a user specified $C > 0$ chosen prior to data collection and for each $j \in [M]$ and $\boldsymbol{i} \in I_{j,k_n}$, set $\alpha_{\boldsymbol{i}, b_n} = \frac{C p_j}{|I_{j,k_n}|}$.

Following this procedure, the total prior concentration on $R_j$ amounts to $Cp_j$. Note, by choosing $C$ prior to data collection, $C$ cannot depend on the potentially growing sample size. So for every $n$, we have that $\sum_{\boldsymbol{i} \in [b_n]^d} \alpha_{\boldsymbol{i}, b_n} = C$ and in particular Assumption 2 is satisfied. Further, by Equations 4 and 9, the Posterior Mean Histogram, $\bar{P}_n$, satisfies that for each $j \in [M]$

$$\bar{P}_n(R_j) = \sum_{\boldsymbol{i} \in I_{j,k_n}} \bar{P}_n(A_{\boldsymbol{i}, b_n}) = \sum_{\boldsymbol{i} \in I_{j,k_n}} \frac{\sum_{t=1}^n \mathbb{I}(Y_t \in A_{\boldsymbol{i}, b_n}) + \frac{Cp_j}{|I_{j,k_n}|}}{n + C} = \frac{\sum_{t=1}^n \mathbb{I}(Y_t \in R_j)}{n} \left( \frac{n}{n+C} \right) + p_j \left( \frac{C}{n+C} \right).$$

This is a convex combination of the empirical proportion and the prior proportion in region $R_j$ for which the weights depend only on the relationship between $C$ and $n$. $C$ should be chosen larger under high confidence prior beliefs and smaller under low confidence prior beliefs. A simple question to elicit $C$ is for the practitioner to ask themselves: for a given small weight $0 < q_0 < 1$, how many samples $n_0$ would need to be available for them to assign this small weight to their prior beliefs and $1 - q_0$ weight to the empirical proportions? Setting $\frac{C}{n_0 + C} = q_0$ and solving yields a choice for $C$.

## 6 Proofs of Auxiliary Lemmas

In this section we prove Lemmas 3, 4 and 5. First we need to state a couple of technical tools.

### 6.1 Technical Tools

The first tool is the multiresolution upper bound on the Wasserstein distance. See Weed & Bach (2019) Section 3 or Singh & Póczos (2018) Appendix A for a good review. Here we use an application of this general result for the metric space $([0,1)^d, \|\cdot\|_2)$.

**Lemma 6.** *(Wasserstein Multiresolution Upper Bound) Let $\mathcal{S}_0 = [0,1)^d$ and for $k \in \mathbb{N}$,*

$$\mathcal{S}_k := \left\{ \left[ \frac{i_1 - 1}{2^k}, \frac{i_1}{2^k} \right) \times \left[ \frac{i_2 - 1}{2^k}, \frac{i_2}{2^k} \right) \times \cdots \times \left[ \frac{i_d - 1}{2^k}, \frac{i_d}{2^k} \right) \ for \ (i_1, i_2, \ldots, i_d) \in [2^k]^d \right\},$$

*If $\mu, \nu$ are probability measures on $[0,1)^d$, then for $p \geq 1$ and $K$ any positive integer,*

$$W_p^p(\mu_1, \mu_2) \leq d^{p/2} \left( \left( \frac{1}{2} \right)^{Kp} + \sum_{k=1}^K \left( \frac{1}{2} \right)^{(k-1)p} \sum_{S \in \mathcal{S}_k} |\mu_1(S) - \mu_2(S)| \right).$$

*Proof.* This is a straightforward application of Proposition 1 of Weed & Bach (2019). $\qquad\square$

The next technical tool is an upper bound on the $L_1$ concentration of a Multinomial distribution around its mean.

**Lemma 7** (Multinomial concentration). *If $(X_1, \ldots, X_k) \sim Multinomial(n, p_1, \ldots, p_k)$ and $Z := \sum_{j=1}^k |X_j - np_j|$, then*

$$\mathbb{E}(Z/n) \leq \sqrt{\frac{k-1}{n}}.$$

*Proof.* Applying Jensen's inequality and then Cauchy-Schwarz

$$\mathbb{E}\left(\frac{Z}{n}\right) \leq \sum_{j=1}^k \sqrt{\mathrm{Var}\left(\frac{X_j}{n}\right)} = \frac{1}{\sqrt{n}} \sum_{j=1}^k \sqrt{p_j(1-p_j)} \leq \frac{1}{\sqrt{n}} \sqrt{\sum_{j=1}^k p_j \sum_{j=1}^k (1-p_j)} = \sqrt{\frac{k-1}{n}}.$$

$$\square$$

The last tool is the concentration of the Dirichlet distribution around its mean in the $L_1$ norm.

**Lemma 8.** *(Dirichlet Concentration) Let $k \in \mathbb{N}$ and $(\pi_1, \pi_2, \ldots, \pi_k) \sim Dirichlet(\alpha_1, \alpha_2, \ldots, \alpha_k)$. Then for $\delta > 0$*

$$\mathbb{P}\left( \sum_{j=1}^{k} |\pi_j - \mathbb{E}(\pi_j)| \geq \frac{(\bar{\alpha})^{-\frac{1}{2}}\sqrt{k}}{\delta} \right) \leq \delta,$$

*where $\bar{\alpha} := \sum_{j=1}^{k} \alpha_j$.*

*Proof.* Basic properties of the Dirichlet distribution give that for $j \in \{1, 2, \ldots, k\}$, $\pi_j \sim \text{Beta}(\alpha_j, \bar{\alpha} - \alpha_j)$. Also, if $X \sim \text{Beta}(\alpha, \beta)$ then $\text{Var}(X) = \alpha\beta/((\alpha + \beta)^2(\alpha + \beta + 1))$. Using these properties, in addition to Jensen's inequality and Cauchy–Schwarz inequality, we have that

$$\begin{aligned}
\mathbb{E}\left( \sum_{j=1}^{k} |\pi_j - \mathbb{E}(\pi_j)| \right) &\leq \sum_{j=1}^{k} \sqrt{\text{Var}(\pi_j)} \\
&= \sum_{j=1}^{k} \sqrt{\frac{\alpha_j(\bar{\alpha} - \alpha_j)}{\bar{\alpha}^2(\bar{\alpha} + 1)}} \\
&\leq (\bar{\alpha})^{-\frac{3}{2}} \sum_{j=1}^{k} \sqrt{\alpha_j(\bar{\alpha} - \alpha_j)} \\
&\leq (\bar{\alpha})^{-\frac{3}{2}} \sqrt{\left( \sum_{j=1}^{k} \alpha_j \right)\left( \sum_{j=1}^{k} \bar{\alpha} - \alpha_j \right)} \\
&= (\bar{\alpha})^{-\frac{3}{2}} \sqrt{\bar{\alpha}(\bar{\alpha}k - \bar{\alpha})} \\
&\leq (\bar{\alpha})^{-\frac{1}{2}} \sqrt{k}.
\end{aligned} \tag{19}$$

By Markov the result follows. $\qquad\square$

### 6.2 Proof of Lemma 3

We now prove Lemma 3.

*Proof of Lemma 3.* By the definition of the partitions $\{\mathcal{S}_k\}_{k \in \mathbb{N}}$ in Lemma 6, the partitions $\mathcal{S}_1, \mathcal{S}_2, \ldots, \mathcal{S}_{K_n}$ are nested in the sense that for $k \in [K_n - 1]$ and $S \in \mathcal{S}_k$, there exists a collections of sets in $\mathcal{S}_{k+1}$ such that $S$ is exactly the union of these sets. In particular the sets $S \in \mathcal{S}_k$ are expressible as unions of sets from $S_{K_n}$. Also using that $b_n = 2^{K_n}$, we have for $k \in \{1, 2, \ldots, K_n\}$ and $S \in \mathcal{S}_k$ a set of indices $I_{S,k,n} \subseteq [b_n]^d$ such that

$$S = \bigcup_{\boldsymbol{j} \in I_{S,k,n}} A_{\boldsymbol{j}, b_n}, \tag{20}$$

where recall $A_{\boldsymbol{j}, b_n}$ is defined in Equation 3. Moreover $\{\cup_{\boldsymbol{j} \in I_{S,k,n}} A_{\boldsymbol{j}, b_n}\}_{S \in \mathcal{S}_k}$ partitions $[0, 1)^d$ and $\{I_{S,k,n}\}_{S \in \mathcal{S}_k}$ partitions $[b_n]^d$. Using this and Lemma 6 (the Wasserstein multiresolution upper bound), we have that

$$\mathbb{E}_{P_0} W_p^p(P_0, \bar{P}_n) \lesssim \left( \frac{1}{2} \right)^{K_n p} + \sum_{k=1}^{K_n} \left( \frac{1}{2} \right)^{(k-1)p} \mathbb{E}_{P_0} \sum_{S \in \mathcal{S}_k} \left| \bar{P}_n\left( \bigcup_{\boldsymbol{j} \in I_{S,k,n}} A_{\boldsymbol{j}, b_n} \right) - P_0(S) \right|. \tag{21}$$

Since the $A_{\boldsymbol{j}, b_n}$ sets are disjoint, for $S \in \mathcal{S}_k$, $\bar{P}_n\left( \bigcup_{\boldsymbol{j} \in I_{S,k,n}} A_{\boldsymbol{j}, b_n} \right) = \sum_{\boldsymbol{j} \in I_{S,k,n}} \bar{P}_n\left( A_{\boldsymbol{j}, b_n} \right)$. By definition of $\bar{P}_n$ (see Equation 9) we thus have

$$\left| \bar{P}_n\left( \bigcup_{\boldsymbol{j} \in I_{S,k,n}} A_{\boldsymbol{j}, b_n} \right) - P_0(S) \right| = \left| \sum_{\boldsymbol{j} \in I_{S,k,n}} \frac{\alpha_{\boldsymbol{j}, b_n} + \sum_{t=1}^{n} \mathbb{I}(Y_t \in A_{\boldsymbol{j}, b_n})}{n + \sum_{\boldsymbol{i} \in [b_n]^d} \alpha_{\boldsymbol{i}, b_n}} - P_0(S) \right|$$

$$
\begin{aligned}
&= \left| \frac{\sum_{\boldsymbol{j} \in I_{S,k,n}} \alpha_{\boldsymbol{j},b_n} + \sum_{t=1}^{n} \mathbb{I}(Y_t \in \bigcup_{\boldsymbol{j} \in I_{S,k,n}} A_{\boldsymbol{j},b_n})}{n + \sum_{\boldsymbol{i} \in [b_n]^d} \alpha_{\boldsymbol{i},b_n}} - P_0(S) \right| \\
&\leq \left| \frac{n}{n + \sum_{\boldsymbol{i} \in [b_n]^d} \alpha_{\boldsymbol{i},b_n}} \frac{\sum_{t=1}^{n} \mathbb{I}(Y_t \in S)}{n} - P_0(S) \right| + \frac{\sum_{\boldsymbol{j} \in I_{S,k,n}} \alpha_{\boldsymbol{j},b_n}}{n + \sum_{\boldsymbol{i} \in [b_n]^d} \alpha_{\boldsymbol{i},b_n}} \\
&\leq \left| \frac{n}{n + \sum_{\boldsymbol{i} \in [b_n]^d} \alpha_{\boldsymbol{i},b_n}} - 1 \right| \frac{\sum_{t=1}^{n} \mathbb{I}(Y_t \in S)}{n} + \left| \frac{\sum_{t=1}^{n} \mathbb{I}(Y_t \in S)}{n} - P_0(S) \right| \quad (22) \\
&\quad + \frac{\sum_{\boldsymbol{j} \in I_{S,k,n}} \alpha_{\boldsymbol{j},b_n}}{n + \sum_{\boldsymbol{i} \in [b_n]^d} \alpha_{\boldsymbol{i},b_n}}.
\end{aligned}
$$

Using this and that $\mathcal{S}_k$ partitions $[0,1)^d$ (and therefore $\frac{1}{n} \sum_{S \in \mathcal{S}_k} \sum_{t=1}^{n} \mathbb{I}(Y_t \in S) = 1$) and Lemma 7 (Multinomial concentration) yields

$$
\mathbb{E}_{P_0} \sum_{S \in \mathcal{S}_k} \left| \bar{P}_n( \bigcup_{\boldsymbol{j} \in I_{S,k,n}} A_{\boldsymbol{j},b_n}) - P_0(S) \right| \lesssim \frac{\sum_{\boldsymbol{j} \in [b_n]^d} \alpha_{\boldsymbol{j},b_n}}{n + \sum_{\boldsymbol{i} \in [b_n]^d} \alpha_{\boldsymbol{i},b_n}} + n^{-\frac{1}{2}} \sqrt{|\mathcal{S}_k|}.
$$

Using this and Equation 21 (and that $|\mathcal{S}_k| = 2^{dk}$ and $K_n \geq \log_2(k_n)$), we have that

$$
\begin{aligned}
\mathbb{E}_{P_0} W_p^p(P_0, \bar{P}_n) &\lesssim \left(\frac{1}{2}\right)^{K_n p} + \frac{\sum_{\boldsymbol{i} \in [b_n]^d} \alpha_{\boldsymbol{i},b_n}}{n + \sum_{\boldsymbol{i} \in [b_n]^d} \alpha_{\boldsymbol{i},b_n}} \sum_{k=1}^{K_n} \left(\frac{1}{2}\right)^{(k-1)p} + n^{-1/2} \sum_{k=1}^{K_n} 2^{-k(p-\frac{d}{2})} \\
&\lesssim k_n^{-p} + \frac{\sum_{\boldsymbol{i} \in [b_n]^d} \alpha_{\boldsymbol{i},b_n}}{n + \sum_{\boldsymbol{i} \in [b_n]^d} \alpha_{\boldsymbol{i},b_n}} + n^{-\frac{1}{2}} \left( \max(1, 2^{K_n(\frac{d}{2}-p)}) \mathbb{I}(d \neq 2p) + K_n \mathbb{I}(d = 2p) \right). \quad (23)
\end{aligned}
$$

Applying Assumptions 1 and 2 now allows us to conclude that $\mathbb{E}_{P_0} W_p^p(P_0, \bar{P}_n) \lesssim n^{-\frac{1}{2}}$ when $d < 2p$, $\lesssim n^{-\frac{p}{d}} \log(n)$ when $d = 2p$ and $\lesssim n^{-\frac{p}{d}}$ when $d > 2p$. The $\lesssim$ arguments of this proof do not depend on $P_0$ (through constants or through eventuality) and so we conclude

$$
\sup_{P_0 \in \mathcal{P}_d} \mathbb{E}_{P_0} W_p^p(P_0, \bar{P}_n) \lesssim \begin{cases} n^{-1/2} & d < 2p. \\ n^{-p/d} \log(n) & d = 2p. \\ n^{-p/d} & d > 2p. \end{cases}
$$

$\square$

### 6.3 Proof of Lemma 4

We now prove Lemma 4.

*Proof of Lemma 4.* We will use the general dual formulation of $W_p$ distance and then apply Mcdiarmid's inequality (McDiarmid et al. (1989); see Proposition 1 of Combes (2024) for the statement of the Standard Mcdiarmid inequality). Towards this end, define for $\mu, \nu \in \mathcal{P}_d$

$$
\overset{*}{W}_p(\mu, \nu) = d^{-1/2} W_p(\mu, \nu). \quad (24)
$$

$\overset{*}{W}_p^p(\mu, \nu)$ is the optimal transport cost between $\mu, \nu$ under the rescaled Polish space $([0,1)^d, d^{-1/2} \| \cdot - \cdot \|_2)$ with optimal transport costs being distance raised to the $p^{th}$ power. Also the diameter of $[0,1)^d$ in the rescaled space is 1 [3]. Thus letting

$$
\mathcal{C}_b := \{ f | f : [0,1)^d \to [0,1], f \text{ continuous w.r.t } d^{-1/2} \| \cdot - \cdot \|_2 \},
$$

---

[3]Rescaling the space to have diameter 1 is not technically nescessary to apply the dual formulation of $W_p$ but we do it here so that we only need to account for the diameter of the true space of interest at the end of the proof

and for $f \in \mathcal{C}_b$ letting $f^c : [0,1)^d \to \mathbb{R}$ be the function satisfying

$$f^c(\boldsymbol{y}) = \sup_{\boldsymbol{x} \in [0,1)^d} \left( f(\boldsymbol{x}) - d^{-p/2} \|\boldsymbol{x} - \boldsymbol{y}\|_2^p \right), \tag{25}$$

the conditions of Weed & Bach (2019) proposition 19 are satisfied and we have for $\mu, \nu \in \mathcal{P}_d$,

$$\overset{*}{W}_p^p(\mu, \nu) = \sup_{f \in \mathcal{C}_b} \mathbb{E}_\mu f - \mathbb{E}_\nu f^c. \tag{26}$$

We now prove the bounded difference inequality nescessary to apply Mcdiarmid's inequality. So let $\boldsymbol{x}_n = (x_1, x_2, \ldots, x_{n-1}, x_n) \in [0,1)^d$ and $\boldsymbol{x}_n' = (x_1, x_2, \ldots, x_{n-1}, x_n') \in [0,1)^d$. Additionally we use the notation $\overset{*}{W}_p(P_0, \bar{P}_n(\boldsymbol{x}))$ to indicate that $\bar{P}_n$ is constructed from $\boldsymbol{x} \in [0,1)^d$, and let $\boldsymbol{j}_x \in [b_n]^d$ and $\boldsymbol{j}_{x'} \in [b_n]^d$ satisfy that $x_n \in A_{\boldsymbol{j}_x, b_n}$ and $x_n' \in A_{\boldsymbol{j}_{x'}, b_n}$ and let $C_{\boldsymbol{j}, b_n} = \sum_{i=1}^n \mathbb{I}(x_i \in A_{\boldsymbol{j}, b_n})$ for $\boldsymbol{j} \in [b_n]^d$. Via equation 26 and expressing expectations as sums over the partition members $A_{\boldsymbol{j}, b_n}$, we thus have that

$$\overset{*}{W}_p \left( \bar{P}_n(\boldsymbol{x}_n), P_0 \right) - \overset{*}{W}_p \left( \bar{P}_n(\boldsymbol{x}_n'), P_0 \right)$$

$$= \sup_{f \in \mathcal{C}_b} \left[ \sum_{\boldsymbol{j} \in [b_n]^d} b_n^d \left( \frac{\alpha_{\boldsymbol{j}, b_n} + C_{\boldsymbol{j}, b_n}}{n + \sum_{\boldsymbol{i} \in [b_n]^d} \alpha_{\boldsymbol{i}, b_n}} \int_{A_{\boldsymbol{j}, b_n}} f(\boldsymbol{x}) d\boldsymbol{x} \right) - \mathbb{E}_{X \sim P_0} \mathbb{I}(X \in A_{\boldsymbol{j}, b_n}) f^c(X) \right] -$$

$$\sup_{f' \in \mathcal{C}_b} \left[ \sum_{\boldsymbol{j} \in [b_n]^d \setminus \{\boldsymbol{j}_x, \boldsymbol{j}_{x'}\}} b_n^d \left( \frac{\alpha_{\boldsymbol{j}, b_n} + C_{\boldsymbol{j}, b_n}}{n + \sum_{\boldsymbol{i} \in [b_n]^d} \alpha_{\boldsymbol{i}, b_n}} \int_{A_{\boldsymbol{j}, b_n}} f'(\boldsymbol{x}) d\boldsymbol{x} \right) - \mathbb{E}_{X \sim P_0} \mathbb{I}(X \in A_{\boldsymbol{j}, b_n}) f'^c(X) + \right. \tag{27}$$

$$b_n^d \left( \frac{a_{\boldsymbol{j}_x, b_n} + C_{\boldsymbol{j}_x, b_n} - \mathbb{I}(\boldsymbol{j}_x \neq \boldsymbol{j}_{x'})}{n + \sum_{\boldsymbol{i} \in [b_n]^d} \alpha_{\boldsymbol{i}, b_n}} \right) \int_{A_{\boldsymbol{j}_x, b_n}} f'(\boldsymbol{x}) d\boldsymbol{x} - \mathbb{E}_{X \sim P_0} \mathbb{I}(X \in A_{\boldsymbol{j}_x, b_n}) f'^c(X) +$$

$$\left. b_n^d \left( \frac{a_{\boldsymbol{j}_{x'}, b_n} + C_{\boldsymbol{j}_{x'}, b_n} + \mathbb{I}(\boldsymbol{j}_x \neq \boldsymbol{j}_{x'})}{n + \sum_{\boldsymbol{i} \in [b_n]^d} \alpha_{\boldsymbol{i}, b_n}} \right) \int_{A_{\boldsymbol{j}_{x'}, b_n}} f'(\boldsymbol{x}) d\boldsymbol{x} - \mathbb{E}_{X \sim P_0} \mathbb{I}(X \in A_{\boldsymbol{j}_{x'}, b_n}) f'^c(X) \right].$$

We write the above equality more succinctly as

$$\overset{*}{W}_p \left( \bar{P}_n(\boldsymbol{x}_n), P_0 \right) - \overset{*}{W}_p \left( \bar{P}_n(\boldsymbol{x}_n'), P_0 \right) = \sup_{f \in \mathcal{C}_b} \left( J_1(f) + J_2(f) \right) - \sup_{f' \in \mathcal{C}_b} \left( J_1(f') + J_3(f') \right), \tag{28}$$

where $J_1 : \mathcal{C}_b \to \mathbb{R}$, $J_2 : \mathcal{C}_b \to \mathbb{R}$, $J_3 : \mathcal{C}_b \to \mathbb{R}$ are defined as

$$J_1(f) := \sum_{\boldsymbol{j} \in [b_n]^d \setminus \{\boldsymbol{j}_x, \boldsymbol{j}_{x'}\}} b_n^d \left( \frac{\alpha_{\boldsymbol{j}, b_n} + C_{\boldsymbol{j}, b_n}}{n + \sum_{\boldsymbol{i} \in [b_n]^d} \alpha_{\boldsymbol{i}, b_n}} \int_{A_{\boldsymbol{j}, b_n}} f(\boldsymbol{x}) d\boldsymbol{x} \right) - \mathbb{E}_{X \sim P_0} \mathbb{I}(X \in A_{\boldsymbol{j}, b_n}) f^c(X),$$

and

$$J_2(f) := \sum_{\boldsymbol{j} \in \{\boldsymbol{j}_x, \boldsymbol{j}_{x'}\}} b_n^d \left( \frac{\alpha_{\boldsymbol{j}, b_n} + C_{\boldsymbol{j}, b_n}}{n + \sum_{\boldsymbol{i} \in [b_n]^d} \alpha_{\boldsymbol{i}, b_n}} \int_{A_{\boldsymbol{j}, b_n}} f(\boldsymbol{x}) d\boldsymbol{x} \right) - \mathbb{E}_{X \sim P_0} \mathbb{I}(X \in A_{\boldsymbol{j}, b_n}) f^c(X),$$

and

$$J_3(f) := b_n^d \left( \frac{a_{\boldsymbol{j}_x, b_n} + C_{\boldsymbol{j}_x, b_n} - \mathbb{I}(\boldsymbol{j}_x \neq \boldsymbol{j}_{x'})}{n + \sum_{\boldsymbol{i} \in [b_n]^d} \alpha_{\boldsymbol{i}, b_n}} \right) \int_{A_{\boldsymbol{j}_x, b_n}} f(\boldsymbol{x}) d\boldsymbol{x} - \mathbb{E}_{X \sim P_0} \mathbb{I}(X \in A_{\boldsymbol{j}_x, b_n}) f'^c(X) +$$

$$b_n^d \left( \frac{a_{\boldsymbol{j}_{x'}, b_n} + C_{\boldsymbol{j}_{x'}, b_n} + \mathbb{I}(\boldsymbol{j}_x \neq \boldsymbol{j}_{x'})}{n + \sum_{\boldsymbol{i} \in [b_n]^d} \alpha_{\boldsymbol{i}, b_n}} \right) \int_{A_{\boldsymbol{j}_{x'}, b_n}} f(\boldsymbol{x}) d\boldsymbol{x} - \mathbb{E}_{X \sim P_0} \mathbb{I}(X \in A_{\boldsymbol{j}_{x'}, b_n}) f'^c(X). \tag{29}$$

Now observe that by adding and subtracting $J_3(f)$

$$\sup_{f \in \mathcal{C}_b} J_1(f) + J_2(f) \leq \sup_{f \in \mathcal{C}_b} J_1(f) + J_3(f) + \sup_{f \in \mathcal{C}_b} J_2(f) - J_3(f). \tag{30}$$

Using this and Equation 28, we have that

$$\overset{*}{W}_p\left(\bar{P}_n(\boldsymbol{x}_n), P_0\right) - \overset{*}{W}_p\left(\bar{P}_n(\boldsymbol{x}'_n), P_0\right) \leq \sup_{f \in \mathcal{C}_b} J_2(f) - J_3(f). \tag{31}$$

In the difference $J_2(f) - J_3(f)$, the expectations involving $f^c$ cancel and thus

$$\begin{aligned}
&\overset{*}{W}_p\left(\bar{P}_n(\boldsymbol{x}_n), P_0\right) - \overset{*}{W}_p\left(\bar{P}_n(\boldsymbol{x}'_n), P_0\right) \\
&\leq \sup_{f \in \mathcal{C}_b} \left[ \left( b_n^d \int_{A_{\boldsymbol{j}_x,b_n}} f(\boldsymbol{x})d\boldsymbol{x} \right) \left( \frac{a_{\boldsymbol{j}_x,b_n} + C_{\boldsymbol{j}_x,b_n}}{n + \sum_{\boldsymbol{i} \in [b_n]^d} \alpha_{\boldsymbol{i},b_n}} - \frac{a_{\boldsymbol{j}_x,b_n} + C_{\boldsymbol{j}_x,b_n} - \mathbb{I}(\boldsymbol{j}_x \neq \boldsymbol{j}_{x'})}{n + \sum_{\boldsymbol{i} \in [b_n]^d} \alpha_{\boldsymbol{i},b_n}} \right) + \right. \\
&\qquad\qquad \left. \left( b_n^d \int_{A_{\boldsymbol{j}_{x'},b_n}} f(\boldsymbol{x})d\boldsymbol{x} \right) \left( \frac{a_{\boldsymbol{j}_{x'},b_n} + C_{\boldsymbol{j}_{x'},b_n}}{n + \sum_{\boldsymbol{i} \in [b_n]^d} \alpha_{\boldsymbol{i},b_n}} - \frac{a_{\boldsymbol{j}_{x'},b_n} + C_{\boldsymbol{j}_{x'},b_n} + \mathbb{I}(\boldsymbol{j}_x \neq \boldsymbol{j}_{x'})}{n + \sum_{\boldsymbol{i} \in [b_n]^d} \alpha_{\boldsymbol{i},b_n}} \right) \right] \\
&= \sup_{f \in \mathcal{C}_b} \frac{b_n^d \mathbb{I}(\boldsymbol{j}_x \neq \boldsymbol{j}_{x'})}{n + \sum_{\boldsymbol{i} \in [b_n]^d} \alpha_{\boldsymbol{i},b_n}} \left( \int_{A_{\boldsymbol{j}_x,b_n}} f(\boldsymbol{x})d\boldsymbol{x} - \int_{A_{\boldsymbol{j}_{x'},b_n}} f(\boldsymbol{x})d\boldsymbol{x} \right) \\
&\leq \frac{1}{n},
\end{aligned} \tag{32}$$

where note in the last inequality we have used that $\mathcal{C}_b$ consists of only non-negative functions bounded above by 1, and that for each $\boldsymbol{j} \in [b_n]^d$, $\int_{A_{\boldsymbol{j},b_n}} d\boldsymbol{x} = \frac{1}{b_n^d}$. By an identical argument, $\overset{*}{W}_p\left(\bar{P}_n(\boldsymbol{x}'_n), P_0\right) - W_p\left(\bar{P}_n(\boldsymbol{x}_n), P_0\right) \leq \frac{1}{n}$ and so we conclude that

$$|\overset{*}{W}_p\left(\bar{P}_n(\boldsymbol{x}'_n), P_0\right) - \overset{*}{W}_p\left(\bar{P}_n(\boldsymbol{x}_n), P_0\right)| \leq \frac{1}{n}.$$

Since $x'_n \in [0,1)^d$ was arbitrary the bounded difference condition in the last coordinate is satisfied with bound $\frac{1}{n}$. The argument for the other $n-1$ coordinates proceeds identically and so we conclude by Mcdiarmid's inequality that for $\epsilon > 0$

$$P_0\left(\overset{*}{W}_p^p\left(\bar{P}_n, P_0\right) - \mathbb{E}_{P_0}\overset{*}{W}_p^p\left(\bar{P}_n, P_0\right) \geq \epsilon \right) \leq \exp(-2n\epsilon^2). \tag{33}$$

Finally recall for all $\mu, \nu \in \mathcal{P}_d$, $W_p(\mu, \nu) = d^{1/2}\overset{*}{W}_p(\mu, \nu)$. Using this and Equation 33 and that $W_p$ is symmetric the lemma follows. □

## 6.4  Proof of Lemma 5

We now prove Lemma 5

*Proof of Lemma 5.* The first inequality we present below upper bounds the $W_p^p$ distance between histograms using the multi-resolution upper bound. So again using Lemma 6 and the sets $I_{S,k,n}$ from Equation 20, we have for $n \in \mathbb{N}, \boldsymbol{\pi}_1, \boldsymbol{\pi}_2 \in \mathcal{S}^{b_n d - 1}$

$$\begin{aligned}
W_p^p(\psi_{b_n}(\boldsymbol{\pi}_1), \psi_{b_n}(\boldsymbol{\pi}_2)) &\leq d^{p/2} \left[ \left(\frac{1}{2}\right)^{K_n p} + \sum_{k=1}^{K_n} \left(\frac{1}{2}\right)^{(k-1)p} \sum_{S \in \mathcal{S}_k} |\psi_{b_n}(\boldsymbol{\pi}_1)(S) - \psi_{b_n}(\boldsymbol{\pi}_2)(S)| \right] \\
&= d^{p/2} \left[ \left(\frac{1}{2}\right)^{K_n p} + \sum_{k=1}^{K_n} \left(\frac{1}{2}\right)^{(k-1)p} \sum_{S \in \mathcal{S}_k} |\psi_{b_n}(\boldsymbol{\pi}_1)(\bigcup_{\boldsymbol{j} \in I_{S,k,n}} A_{\boldsymbol{j},b_n}) - \psi_{b_n}(\boldsymbol{\pi}_2)(\bigcup_{\boldsymbol{j} \in I_{S,k,n}} A_{\boldsymbol{j},b_n})| \right] \\
&= d^{p/2} \left[ \left(\frac{1}{2}\right)^{K_n p} + \sum_{k=1}^{K_n} \left(\frac{1}{2}\right)^{(k-1)p} \sum_{S \in \mathcal{S}_k} \left| \sum_{\boldsymbol{j} \in I_{S,k,n}} \pi_{1\boldsymbol{j}} - \sum_{\boldsymbol{j} \in I_{S,k,n}} \pi_{2\boldsymbol{j}} \right| \right].
\end{aligned} \tag{34}$$

Also, for $\boldsymbol{\pi}_1 \in \mathcal{S}^{b_n d - 1}$ and $k \in [K_n]$, define

$$V_n(\boldsymbol{\pi}_1, k) := \sum_{S \in \mathcal{S}_k} \left| \sum_{\boldsymbol{j} \in I_{S,k,n}} \pi_{1\boldsymbol{j}} - \sum_{\boldsymbol{j} \in I_{S,k,n}} \mathbb{E}_{z_n^*}(\pi_{\boldsymbol{j}} | Y_1, \ldots, Y_n) \right|. \tag{35}$$

Using Equation 34, the definition in Equation 35, the preimage form of $\Pi_n$ (Equation 7), the definition of $\bar{P}_n$ (Equation 9), the definition of $z_n^*$ (the posterior measure over the simplex $\mathcal{S}^{b_n d - 1}$), and that by Assumption 1 and the definition of $\tau_n(d, p)$ in Equation 12, $2^{-K_n p} = o(\tau_n^p(d, p))$ (as $n \to \infty$) we have that almost surely under $P_0$ and eventually in $n$ and for each $d \in \mathbb{N}, p \geq 1$

$$
\begin{aligned}
&\Pi_n(P \in \mathcal{P}_d : W_p(P, \bar{P}_n) \geq \tau_n(d, p)) \\
&= z_n^*(\boldsymbol{\pi}_1 \in \mathcal{S}^{b_n d - 1} : W_p^p(\psi_{b_n}(\boldsymbol{\pi}_1), \psi_{b_n}(\mathbb{E}_{z_n^*}(\boldsymbol{\pi} | Y_1, \ldots, Y_n))) \geq \tau_n^p(d, p)) \\
&\leq z_n^*\left( \boldsymbol{\pi}_1 \in \mathcal{S}^{b_n d - 1} : \frac{1}{2^{K_n p}} + \sum_{k=1}^{K_n} \frac{1}{2^{(k-1)p}} V_n(\boldsymbol{\pi}_1, k) \geq d^{-p/2} \tau_n^p(d, p) \right) \\
&\leq z_n^*\left( \boldsymbol{\pi}_1 \in \mathcal{S}^{b_n d - 1} : \sum_{k=1}^{K_n} \frac{1}{2^{(k-1)p}} V_n(\boldsymbol{\pi}_1, k) \geq \frac{1}{2} d^{-p/2} \tau_n^p(d, p) \right) \\
&\leq z_n^*\left( \boldsymbol{\pi}_1 \in \mathcal{S}^{b_n d - 1} : \sum_{k=1}^{K_n} \frac{1}{2^{kp}} V_n(\boldsymbol{\pi}_1, k) \geq 2^{p-1} d^{-p/2} \tau_n^p(d, p) \right).
\end{aligned}
\tag{36}
$$

We will now derive an upper bound on $\sup_{\boldsymbol{\pi}_1 \in \mathcal{S}^{b_n d - 1}} \max_{j \in [K_n]} V_n(\boldsymbol{\pi}_1, k)$ which, if it holds, ensures the event inside the probability of the last line of Equation 36 must be false. We will use this upper bound and the union bound to control from above the probability in the last line of Equation 36. Regarding the upper bound on $\sup_{\boldsymbol{\pi}_1 \in \mathcal{S}^{b_n d - 1}} \max_{j \in [K_n]} V_n(\boldsymbol{\pi}_1, k)$, note that

$$\sum_{k=1}^{K_n} 2^{-kp} \left( \frac{\log^\gamma(n) 2^{\frac{dk}{2}}}{\sqrt{n + \sum_{\boldsymbol{i} \in [b_n]^d} \alpha_{\boldsymbol{i}, b_n}}} \right) = \frac{\log^\gamma(n)}{\sqrt{n + \sum_{\boldsymbol{i} \in [b_n]^d} \alpha_{\boldsymbol{i}, b_n}}} \sum_{k=1}^{K_n} 2^{-k(p - \frac{d}{2})} \lesssim \tau_n^p(d, p). \tag{37}$$

To see the $\lesssim$ in Equation 37, observe that by Assumption 2, the total prior concentration is dominated by $n$ and therefore the term in front of the summand on LHS of $\lesssim$ is $\asymp \frac{\log^\gamma(n)}{\sqrt{n}}$. Thus by definition of $\tau_n^p(d, p)$ (Equation 12), this is sufficient to conclude $LHS \lesssim \tau_n^p(d, p)$ in the $d < 2p$ case. In the $d = 2p$ case the sum contributes a factor $\log(n)$ to LHS and so again $LHS \lesssim \tau_n^p(d, p)$. In the $d > 2p$ case, the sum contributes an asymptotic factor $2^{K_n(\frac{d}{2} - p)} \asymp k_n^{\frac{d}{2} - p} = n^{\frac{1}{2} - \frac{p}{d}}$ to LHS so that $LHS \asymp \log^\gamma(n) n^{-\frac{p}{d}}$ and so again $LHS \lesssim \tau_n^p(d, p)$. By Equation 37, for each $d \in \mathbb{N}, p \geq 1$, we set $C_1(d, p)$ sufficiently large so that eventually in $n$

$$\sum_{k=1}^{K_n} 2^{-kp} \left( \frac{\log^\gamma(n) 2^{\frac{dk}{2}}}{\sqrt{n + \sum_{\boldsymbol{i} \in [b_n]^d} \alpha_{\boldsymbol{i}, b_n}}} \right) < 2^{p-1} d^{-p/2} \tau_n^p(d, p). \tag{38}$$

The upper bound on $\sup_{\boldsymbol{\pi}_1 \in \mathcal{S}^{b_n d - 1}} \max_{j \in [K_n]} V_n(\boldsymbol{\pi}_1, k)$ we are seeking is thus

$$\frac{\log^\gamma(n) 2^{\frac{dk}{2}}}{\sqrt{n + \sum_{\boldsymbol{i} \in [b_n]^d} \alpha_{\boldsymbol{i}, b_n}}}.$$

Thus with $C_1(d, p)$ sufficiently large so that Equation 38 holds, we have that eventually in $n$ and almost surely under $P_0$

$$z_n^*\left( \boldsymbol{\pi}_1 \in \mathcal{S}^{b_n d - 1} : \sum_{k=1}^{K_n} \frac{1}{2^{kp}} V_n(\boldsymbol{\pi}_1, k) \geq 2^{p-1} d^{-p/2} \tau_n^p(d, p) \right)$$

$$\leq z_n^* \left( \boldsymbol{\pi}_1 \in \mathcal{S}^{b_n d - 1} : \exists k \in \{1, 2, \ldots, K_n\} \text{ s.t } V_n(\boldsymbol{\pi}_1, k) > \log^\gamma(n) \sqrt{\frac{2^{dk}}{n + \sum_{\boldsymbol{j} \in [b_n]^d} \alpha_{\boldsymbol{j}, b_n}}} \right)$$

$$\leq \sum_{k=1}^{K_n} z_n^* \left( \boldsymbol{\pi}_1 \in \mathcal{S}^{b_n d - 1} : V_n(\boldsymbol{\pi}_1, k) > \log^\gamma(n) \sqrt{\frac{2^{dk}}{n + \sum_{\boldsymbol{j} \in [b_n]^d} \alpha_{\boldsymbol{j}, b_n}}} \right), \tag{39}$$

where in the last line we have used the union bound. Now recall $\{I_{S,k,n}\}_{S \in \mathcal{S}_k}$ partitions $[b_n]^d$ for $k \in \{1, 2, \ldots, K_n\}$. In particular, since $z_n^* = \text{Dirichlet}(\cdot | \{\alpha_{\boldsymbol{j}, b_n}^*\}_{\boldsymbol{j} \in [b_n]^d})$, under $z_n^*$, $\{\sum_{\boldsymbol{j} \in I_{S,k,n}} \pi_{\boldsymbol{j}}\}_{S \in \mathcal{S}_k} \sim$ $\text{Dirichlet}(\{\sum_{\boldsymbol{j} \in I_{S,k,n}} \alpha_{\boldsymbol{j}, b_n}^*\}_{S \in \mathcal{S}_k})$. Moreover, $\sum_{S \in \mathcal{S}_k} \sum_{\boldsymbol{j} \in I_{S,k,n}} \alpha_{\boldsymbol{j}, b_n}^* = \sum_{\boldsymbol{j} \in [b_n]^d} \alpha_{\boldsymbol{j}, b_n}^* \overset{a.s}{=} n + \sum_{\boldsymbol{j} \in [b_n]^d} \alpha_{\boldsymbol{j}, b_n}$. Finally note that by definition of $\mathcal{S}_k$, $|\mathcal{S}_k| = 2^{dk}$. So for $n \in \mathbb{N}$ and $k \in \{1, 2, \ldots, K_n\}$ applying Dirichlet concentration of measure Lemma 8 with $\delta := \log^{-\gamma}(n)$, we have that for $C_1(d, p)$ sufficiently large, eventually in $n$ and almost surely under $P_0$

$$\sum_{k=1}^{K_n} z_n^* \left( \boldsymbol{\pi}_1 \in \mathcal{S}^{b_n d - 1} : V_n(\boldsymbol{\pi}_1, k) > \log^\gamma(n) \sqrt{\frac{2^{dk}}{n + \sum_{\boldsymbol{j} \in [b_n]^d} \alpha_{\boldsymbol{j}, b_n}}} \right) \leq K_n \log^{-\gamma}(n). \tag{40}$$

Finally note $K_n = \lceil \log(k_n) \rceil \leq \log(k_n) + 1 \leq 2 \log(n)$ and the asymptotic arguments of this proof do not depend on $P_0$ either through constants or eventuality. Thus by Equations 36, 39 and 40, we have that for each $d \in \mathbb{N}$ and $p \geq 1$ and $C_1(d, p)$ sufficiently large, and an $N(d, p)$ (depending only on $d, p$ but not on $P_0$), and $n \geq N$

$$\Pi_n(P \in \mathcal{P}_d : W_p(P, \bar{P}_n) \geq \tau_n(d, p)) \leq 2 \log^{1-\gamma}(n) \tag{41}$$

almost surely under $P_0$. $\qquad \square$

# 7 Conclusions

In this work we obtained minimax optimal PCRs for unconstrained distribution estimation on $[0, 1]^d$ underneath the Wasserstein-$p$ distances for every data dimension $d$. To the best of our knowledge these are the first PCRs achieving minimaxity for every problem dimension $d$ under $W_p, p \geq 1$ distance. Our proof technique avoids verifying a Kullback-Liebler prior support condition by using conjugacy and a direct analysis of the posterior distribution.

These results may be useful to practitioners needing to estimate a distribution underneath a Wasserstein distance when they have some knowledge prior to data collection about the shape of the distribution they are estimating, intend to encode this through a prior distribution to potentially achieve increased accuracy at low sample sizes, and yet simultaneously require a guarantee of precision at large sample sizes that is robust to inaccurate prior selection.

An important area for future work is to determine whether for high dimensional data, Bayesian models can adaptively achieve minimax optimal PCRs underneath Wasserstein-$p$ distances in constrained distribution estimation settings where it is safe to assume that the distribution to be estimated is of low entropy or has a smooth density.

# 8 Appendix

## 8.1 Proofs of Lemmas Regarding Minimax-Conscious PCRs

In this section we prove the lemmas that relate PCRs to minimax rates. The first proof is inspired by Ghosal et al. (2000) Theorem 2.5.

*Proof of Lemma 1.* Define $H_n : \mathcal{P}_d \to [0, 1]$ as

$$H_n(Q) := \Pi_n(P \in \mathcal{P}_d : \tilde{d}(P, Q) < \epsilon_n)$$

and

$$Z_n := \sup_{Q \in \mathcal{P}_d} H_n(Q)$$

and define the $n^{th}$ estimator $\hat{P}_n$ to satisfy $H_n(\hat{P}_n) > Z_n - \frac{1}{6}$; recall $\Pi_n$ is a function of the sample $Y_1, \ldots, Y_n$ and therefore so is $\hat{P}_n$. By the lemma assumption, and since $z_n < \frac{1}{3}$ eventually, there exists an $N$ such that for $n \geq N$ such that for every $P_0 \in \mathcal{P}_d$

$$P_0\left(H_n(P_0) > 2/3\right) \geq 1 - \delta_n \tag{42}$$

And since for $P_0 \in \mathcal{P}_d$, the event $H_n(P_0) > \frac{2}{3}$ implies $Z_n > \frac{2}{3}$, and in particular $H_n(\hat{P}_n) > \frac{1}{2}$, we also have that for $n \geq N$ and every $P_0 \in \mathcal{P}_d$

$$P_0\left(\left[H_n(\hat{P}_n) > \frac{1}{2}\right] \cap \left[H_n(P_0) > \frac{2}{3}\right]\right) \geq 1 - \delta_n \tag{43}$$

Now note that by definition of $H_n$ and since $\Pi_n$ is a probability measure, for every $P_0$,

$$\left[H_n(\hat{P}_n) > \frac{1}{2}\right] \cap \left[H_n(P_0) > \frac{2}{3}\right] \subseteq \left[\exists Q \in \mathcal{P}_d : \max(\tilde{d}(P_0, Q), \tilde{d}(\hat{P}_n, Q)) < \epsilon_n\right]$$

Using this and Equation 43 and triangle inequality, for $n \geq N$ and each $P_0 \in \mathcal{P}_d$,

$$P_0(\tilde{d}(P_0, \hat{P}_n) < 2\epsilon_n) \geq 1 - \delta_n \tag{44}$$

Using this and that $\tilde{d}$ is bounded above on $\mathcal{P}_d$ and that $\delta_n \lesssim \epsilon_n$ by assumption, we have that for $n \geq N$

$$\sup_{P_0 \in \mathcal{P}_d} \mathbb{E}_{P_0} d(P_0, \hat{P}_n) \lesssim \epsilon_n + \delta_n \lesssim \epsilon_n \tag{45}$$

Thus the lemma statement now follows by the definition of the minimax rate. □

*proof of Lemma 2.* Let $P_0 \in \mathcal{P}_d$. By the lemma assumption, we have that

$$\sum_{n=1}^{\infty} P_0\left(\Pi_n(P \in \mathcal{P}_d : \tilde{d}(P, P_0) \geq \epsilon_n) > z_n\right) \leq \sum_{n=1}^{\infty} \delta_n < \infty \tag{46}$$

By the Borel-Cantelli Lemma we conclude that with probability 1 under $P_0$, it is eventually true that

$$\Pi_n(P \in \mathcal{P}_d : \tilde{d}(P, P_0) \geq \epsilon_n) \leq z_n$$

Since $z_n \to 0$, we conclude that almost surely under $P_0$

$$\Pi_n(P \in \mathcal{P}_d : \tilde{d}(P, P_0) \geq \epsilon_n) \to 0$$

□

In the introduction we claim that an almost sure PCR holding uniformily over a finite distribution class cannot decay faster than the minimax rate for that class. While we do not actually use this claim in our work, for completeness, a proof of this claim is provided below. This lemma is also inspired by Ghosal et al. (2000) Lemma 2.5.

**Lemma 9.** *Let $(\mathcal{C}, \tilde{d})$ be a finite metric space of probability distributions, and $\Pi_n$ a sequence of posterior distributions over $\mathcal{C}$. Further suppose for every $P_0 \in \mathcal{C}$, whenever $Y_1, \ldots, Y_n, \ldots \overset{iid}{\sim} P_0$,*

$$\Pi_n(P \in \mathcal{C} : \tilde{d}(P_0, P) \geq \epsilon_n) \to 0 \ P_0 \ almost \ surely.$$

*Then there exists a sequence of estimators $\hat{P}_n \in \mathcal{C}$, such that*

$$\max_{P_0 \in \mathcal{C}} \mathbb{E}_{P_0} \tilde{d}(P_0, \hat{P}_n) \leq 2\epsilon_n$$

*for sufficiently large n. In particular if $z_n$ satisfies*

$$\inf_{\tilde{P}} \max_{P_0 \in \mathcal{C}} \mathbb{E}_{P_0} \tilde{d}(P_0, \tilde{P}) \gtrsim z_n$$

*where the* inf *is taken over estimators $\tilde{P}$ based on n observations, then $\epsilon_n \gtrsim z_n$.*

*Proof.* Define

$$H_n : \mathcal{C} \to [0, 1]$$

as

$$H_n(Q) := \Pi_n(P \in \mathcal{C} : \tilde{d}(P, Q) < \epsilon_n)$$

and

$$Z_n := \max_{Q \in \mathcal{C}} H_n(Q)$$

and let $\hat{P}_n = \arg\max_{Q \in \mathcal{C}} H_n(Q)$. For $P_0 \in \mathcal{C}$, since

$$\Pi_n(P \in \mathcal{C} : \tilde{d}(P_0, P) \geq \epsilon_n) \to 0 \ P_0 \text{ almost surely,}$$

we have with probability 1 under $P_0$ there is an $N(P_0)$ such that for $n \geq N(P_0)$

$$H_n(P_0) > \frac{1}{2}$$

In particular $H_n(\hat{P}_n) > \frac{1}{2}$ as well. Since $\Pi_n$ is a probability measure and with probability 1 under $P_0$, when $n \geq N(P_0)$, $\min(H_n(\hat{P}_n), H_n(P_0)) > \frac{1}{2}$, we have that

$$\{P \in \mathcal{C} : \tilde{d}(P, \hat{P}_n) < \epsilon_n\} \cap \{P \in \mathcal{C} : \tilde{d}(P, P_0) < \epsilon_n\} \neq \emptyset$$

with probability 1 under $P_0$ for $n \geq N(P_0)$. Letting $Q$ be any point common to both these sets, we have that

$$\tilde{d}(P_0, \hat{P}_n) \leq \tilde{d}(P_0, Q) + \tilde{d}(Q, \hat{P}_n) \leq 2\epsilon_n$$

with probability 1 under $P_0$ for $n \geq N(P_0)$. In particular, $\mathbb{E}_{p_0} \tilde{d}(P_0, \hat{P}_n) \leq 2\epsilon_n$ for $n \geq N(P_0)$. This argument holds for all $P_0 \in \mathcal{C}$, so letting $N = \max_{P \in \mathcal{C}} N(P)$, we conclude that for $n \geq N$

$$\max_{P_0 \in \mathcal{C}} \mathbb{E}_{P_0} \tilde{d}(P_0, \hat{P}_n) \leq 2\epsilon_n$$

$\square$

# 9 Acknowledgements

This work was supported by contracts DOE DE-SC0021015, NSF CCF-2115677, and the Laboratory Directed Research and Development program at Sandia National Laboratories, a multimission laboratory managed and operated by National Technology and Engineering Solutions of Sandia, LLC, a wholly-owned subsidiary of Honeywell International, Inc., for both the U.S. Department of Energy's National Nuclear Security Administration under contract DE-NA0003525. This paper describes objective technical results and analysis. Any subjective views or opinions that might be expressed in the paper do not necessarily represent the views of the U.S. Department of Energy or the United States Government. The authors further acknowledge and thank both Jeff Phillips and the reviewers for helpful conversations.

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
