# OpenReview forum: "Minimax Posterior Contraction Rates for Unconstrained Distribution Estimation on $[0,1]^d$ under Wasserstein Distance"
_TMLR — Accepted by TMLR_

### Review · Reviewer_iaVK · 2024-10-22

**Summary Of Contributions:**

This paper studies the nonparametric estimation of a distribution on $[0,1]^d$ ($d\geq 1$)  in the Wasserstein distance $W_p$ $(p\geq 1)$ given $n$ iid samples. The authors use a Bayesian framework: the proposed estimator is the posterior under a Bayesian Histogram model, with bin weights drawn from a Dirichlet distribution ; whose parameters should be specified by the practitioner, for each $n$. This paper shows that this estimator achieves the minimax rates, in the sense that the upper bounds on the asymptotic estimation rate in $n$ shown in the main result (Theorem 1) match the known lower bounds up to logarithmic terms (due to Singh & Póczos (2018)). The upper bound are formulated in the form of PCR (Posterior contraction rates) and the proof technique relies on the multiresolution upper-bound technique  (due to Weed & Bach).

**Audience:**

Yes

**Broader Impact Concerns:**

None.

**Claims And Evidence:**

Yes

**Requested Changes:**

- Some of the vocabulary and facts from Bayesian statistics would deserve a bit more pedagogy to the benefit of the general TMLR reader. For instance: p3. “Due to conjugacy, it is straightforward to show that” (include reference, explain which distributions are conjugate to each other). Also more details on the motivation behind the PCR formalism and its exact definition would be welcome (e.g. the "slowly increasing" sequence $M_n$ in p1 was quite mysterious to me).
- The minimax rate in expectation (Eq.(11)) and the PCR rates of Thm. 1 are not stated in the same formalism. Can you explain why they are indeed directly comparable? In particular, does Eq. (11) indeed give a bound on the fastest PCR rate achievable? This sounds reasonable but deserves justification, in particular because the minimax claim is in the title of the manuscript.

**Strengths And Weaknesses:**

Strengths:
- the paper is mathematically rigorous, concise and straight to the point. Its contributions are clearly stated and its proof well organized.
- I do not know about the PCR formalism, and I could understand it from this paper alone; I learnt interesting facts and tools by reading this manuscript.
- The proposed estimator is quite natural (a Bayesian Histogram with adaptive bin size)  and it is potentially useful for the literature to have an analysis of this estimator for the Wasserstein distance, in arbitrary dimension. It is also interesting to know that it is minimax (and the conditions on the prior parameters that are optimal).

Weaknesses:
- It is not clear to me what is the advantage of the proposed estimator compared to the "empirical distribution" estimator which already achieves the minimax rates (up to log-terms). From the point of view of the Wasserstein metric, there does not appear to be any advantage of this estimator. (But their must be other, stronger, metrics on the space of distributions that favor this more regular estimator).
- I had difficulties understanding what was precisely done in prior works from the literature review in Section 1. This section seems more concerned about explaining what was what *not* done in prior work (each mentioned paper is quickly followed by "... but they do not ..."); this matters, but a more "positive" overview of the literature would be useful to non-expert readers.

---

> ### Author Response · Authors · 2024-11-27
> **Response to Reviewer iaVK**
>
> We again want to thank you for the time and effort you put into reviewing our paper and providing thoughtful feedback. Your comments regarding connecting PCR results to minimaxity were especially helpful. Here we will address the weaknesses and requested changes that you mention.
>
> **Weakness 1**
>
>
> While we believe the main contribution of our work is in advancing posterior contraction theory to achieve minimax optimal PCRs underneath Wasserstein distances under arbitrary dimensions, note that the Bayesian histogram provides the practitioner with a natural mechanism for encoding prior knowledge about the unknown distribution into the analysis; the empirical measure does not. By showing the asymptotic minimax optimality of the Bayesian histogram, we have demonstrated a robustness property of this model; namely the method is (via the minimax criteria) competitive with the empirical measure at large sample sizes but also, subject to the constraints of assumption 2, can support a practitioners correct or incorrect prior knowledge about the unknown distribution $P_0$, which may improve accuracy when the sample size is smaller and the prior knowledge provided is correct.
>
> We have now included a new section (“Using Prior Information” — section 5) that explains the kinds of prior beliefs that can be incorporated into this model, and the steps a practitioner would need to take to select a prior under such beliefs.
>
> **Weakness 2**
>
> We have now added additional background content to give the TMLR audience a better survey of important results in PCR theory. Specifically, we have added three references to successful applications of the Ghosal, Ghosh, and van der Vaart technique for proving posterior contraction for distribution estimation as well as additional explanation for the contributions regarding estimation of mixing distributions. We have also generally made the introduction more positive, by adding a line emphasizing the achievements made in PCR theory under Wasserstein distance in 1 dimension (by Chae) and another on the achievements in developing PCRs when Bayes Theorem is not available (by Camerlenghi).
>
> **Requested Change 1**
>
> We have added a reference for the conjugacy we refer to in section 2.2 and now mention the distributions involved in the conjugate pair. We address your second statement in our response to the next requested change.
>
> **Requested Change 2**
>
> We especially thank the reviewer for this question. In an oversight on our part, it is not necessarily the case that an in probability PCR holding over a distribution class must decay no faster than the minimax rate. One needs a stronger criteria and this had led us to significantly strengthen our result.
>
> In the introduction we now first introduce PCRs that hold almost surely. The almost sure PCR definition is often referred to in the Bayesian Nonparametrics community as a “strong” PCR - it is formally introduced in definition 8.1 of Ghosal and van der Vaart Fundamentals of Nonparametric Bayesian Inference. We then provide a frequentist motivation for the “strong” PCR definition in the introduction, which is that when it holds uniformly over a distribution class, this implies the existence of an estimator derived from the posterior distribution that attains the same convergence rate uniformly over the class (at least when the distribution class is finite) — see the newly added appendix Lemma 9.
>
> In our work, the distribution class is not necessarily finite. To address this, we have added a new section, section 3, which introduces a slightly stronger notion (see definition 1) than the “strong” PCR. We call this a minimax-conscious PCR, and in this section we prove two properties about it. The first (lemma 1) is that a minimax-conscious PCR can never decay at a faster rate than the minimax rate because it ensures the existence of an estimator derived from the posterior distribution achieving the minimax rate over the distribution class. The second (lemma 2) shows that when a minimax-conscious PCR holds over a distribution class, the same PCR holds in the “strong” sense for every distribution in the class. The proofs of lemmas 1 and 2 are now included in the appendix.
>
> Theorem 1 has thus been adjusted to prove a minimax-conscious PCR. Note also that to get this stronger result, we needed to prove a concentration of measure property for the $W_p$ distance between the posterior mean histogram and the truth around its mean. This new lemma is lemma 4.

---

> > ### Comment · Reviewer_iaVK · 2024-11-28
> > **Ready for publication**
> >
> > Thanks for taking into account our reviews. I'm glad that making the connection with the lower bounds rigorous led to a clarification of the relationship between various notions of minimax optimality. I think that the paper is ready for publication!

---

### Review · Reviewer_9k4x · 2024-10-27

**Summary Of Contributions:**

This paper derives a minimax posterior constraction rate for the estimation of a distribution $P_0$ on $[0,1]^d$ under the $p$-Wasserstein distance for $p\ge 1$. This rate is derived using Bayesian histograms.

**Audience:**

Yes

**Claims And Evidence:**

Yes

**Requested Changes:**

To be honest, I am not an expert in statistics, and even less in Posterior Contraction Rates, so I will try to give some hints on what could be improved. But I probably did not understand a lot of things.

While the content of the paper is technical, it might be possible to make it more friendly to read for a broader audience, e.g. by adding more background on the previous related results and what have been done. For instance, the beginning of the second page describes related works, but is very hard to follow.

It would also help the reader to have a concrete example of an estimator. I personally did not understand well how to estimate in practice the distribution P using this method.

Please improve the abstract. The two first sentences seem basically to be the sames.

Also using the Wasserstein-$v$ metric does not seem very standard, as usually $p$ is used.

Typo:
- "practicioners" in Conclusion.

**Strengths And Weaknesses:**

**Strengths:**

- The rate derived under no assumption on $P_0$ seems new and optimal
- The Bayesian histogram technique seems to be original

**Weaknesses:**

- The paper is generally not very well written and hard to follow.
- It is also very technical. While it might interest some people in Optimal Transport, it is a paper which might interest more a statistic audience.
- There is no experiment showing the rate.

---

> ### Author Response · Authors · 2024-11-27
> **Response to Reviewer 9k4x**
>
> We again want to thank you for the time and effort you put into reviewing our paper and providing thoughtful feedback. Here we will address the requested changes that you mention.
>
> **Requested Change 1**
> (This is regarding your change request that starts with the words "While the content of the paper is technical...")
>
> On the second page, we have now added information on several influential works (Scricciolo 2007, Kruijer \& van der Vaart 2008, Shen et al 2013) that have made contributions in proving PCRs for various different instances of the distribution estimation problem and described how these settings are different from the one we consider. We have also included more background on the prior works that we originally referenced, specifically highlighting the accomplishments of (Chae 2021, Camerlenghi 2022), and explain precisely what the inferential goal is in works that estimate a mixing distribution.
>
> **Requested Change 2**
> (This is regarding your change request that starts with the words "It would also help the reader...")
>
> While we believe the main contribution of our work is in advancing posterior contraction theory to achieve minimax optimal PCRs underneath Wasserstein distances under arbitrary dimensions, in practice one would use the posterior mean histogram estimator. Its definition is provided in section 2.2 equation 9, and it is analyzed in lemma 3 and achieves minimax optimality. To help emphasize this point, we have also included a paragraph at the end of section 4 (right after the proof of theorem 1), which states "In practice, the Posterior Mean Histogram, introduced in equation 9 and shown to achieve the same rates for estimation as the posterior itself in Lemma 3, can be used as the functional point estimator for the unknown distribution."
>
> **Requested Change 3**
> (This is regarding your change request which starts with the words "Please improve the abstract.")
>
> We have updated the second sentence to be more succinct. Thank you for pointing this out.
>
> **Requested Change 4**
> (This is regarding your change request which starts with the words "Also using the Wasserstein ...")
>
> We have changed $v$ to $p$.

---

> > ### Comment · Reviewer_9k4x · 2024-11-29
> >
> > Thank you for addressing my concerns, and for revising the paper. It is now much clearer to read.

---

### Review · Reviewer_feWi · 2024-11-24

**Summary Of Contributions:**

The authors derive minimax optimal posterior contraction rates (PCRs) under quite general conditions and for any number of dimensions or Wasserstein-$u$ metric (that is Theorem 1, which is the main contribution of the paper). The authors consider the unconstrained estimation case where we wish to approximate some $P_0$ probability measure on $[0,1]^d$ given a set of samples. No other assumptions are placed on $P_0$. To set up a posterior for study, the authors use a Bayesian Histogram model that partitions $[0,1]^d$ into equal area squares, where a sequence $k_n$ determines the number of bins and n denotes the number of data points. A multinomial likelihood and Dirichlet prior combination results in a Dirichlet posterior (due to conjugacy). Then the authors induce concentration assumptions on the prior and specific rates of increase for $k_n$ and with those (and the conjugacy) show that the PCRs of the proposed model are, up to log terms, the minimax rates from Singh and Póczos (2018).

**Audience:**

Yes

**Broader Impact Concerns:**

No ethical concerns.

**Claims And Evidence:**

Yes

**Requested Changes:**

# Clarifications for acceptance

I would appreciate clarification of these points by the authors, either in discussion and/or in edits of the paper.

While I could follow most of the proofs, the final proof of Lemma 2 takes some effort. I suggest:

1. When constructing the upper bounds in Equation (23), most terms don't change in the long expression. Consider describing before Equation (23) what is going to happen or otherwise refactoring the presentation of the bounds by assigning terms that don't change to some shorthand (see next note).

2. Similarly, around Equation (26), it is difficult to follow the steps leading to the final upper bound. One idea would be to use a temporary expression, e.g.,

$$V(\ldots):=\sum_{S\in S_k}\left | \sum_{j\in I_{S,k,n}}\ldots\right |,$$

to clean up the wider expression (as that part does not change during the derivation of the bounds) and focus the reader on what actually changes. The authors have the space to even attach a number to this expression to make it easy to refer to. This expression could then be reused to simplify Equation (27) as well.

Generally, I suggest rewording a bit Lemma 1. and Lemma 2. to make them easier to parse.

# A few suggestions that would help with presentation, but are not essential for acceptance

1. Clarify whether the expectation under Equation (12) should be $E_{P_0}$ or $E_{p_0}$ (the difference is the capitalization of $P_0$). Same for all following expectations in Proof of Theorem 1. This appears to be a typo.
2. Capitalize all proper nouns in text, e.g., "equation 1" → "Equation 1", "proof of lemma 1" → "proof of Lemma 1", etc.
3. Define what $a_n\lesssim b_n$ means in Section 2.1 to help readers not immediately familiar with the notation.
4. In Equation (14), consider whether cancelling the terms on the right of the $\lesssim$ symbol, e.g., $n^{-1/2u}/(n^{-1/2u}\log^{\gamma/u}(n))=1/(\log^{\gamma/u}(n))$, etc., would help the reader more quickly see the limit behavior.
5. Under Equation (18), refer to the equation where $A_{j,b_n}$ are defined, e.g., "where $A_{j,b_n}$ as defined in Equation (3)."
6. The multi-resolution lemma (Lemma 3) does not define the K constant; a statement like "Let K be a positive integer" would be sufficient.
7. In Section 4.2, after "Proof of Lemma 1", the authors state "Due to the nesting of the dyadic models". I appreciate that this is a standard term for mathematicians, but it probably would be helpful for the reader if the authors explicitly stated which are those models.

**Strengths And Weaknesses:**

### Strengths

-  Theorem 1 does not put constraints on $P_0$  or strong assumptions on the shape of the prior distribution, and I find Assumption 2 reasonable.
- I think readers would find the both the main result (Theorem 1) and the analysis in the two supporting lemmas (Lemma 1 and Lemma 2) interesting.
- The authors communicate a clear strategy for proving the result through the text.


### Weaknesses
- Perhaps because of page limitations, some of the upper bound derivations in the proofs of Lemma 1 and Lemma 2 carry around a lot of symbols and indexes, which makes it a bit hard to follow the steps taken. I think it’s important to revise those a bit to make it easier for the readers, as those two lemmas reveal a lot about Theorem 1.

---

> ### Author Response · Authors · 2024-11-27
> **Response to Reviewer feWi**
>
> We again want to thank you for the time and effort you put into reviewing our paper and providing  feedback regarding clarity of exposition. Here we will address the requested changes that you mention.
>
> **Requested Change 1 and Requested Change 2**
>
> To increase readability in Lemma 2 (which is now Lemma 5) we assign the double sum its own notation — $V_n(\pi,k)$ — see equation 35. We have also added a written explanation before equation 37 (which was equation 24) to provide the reader with immediate guidance as to how equation 37 is used later in the proof — so the reader has better foresight into how equation 36 and 37 (which were equations 23 and 24) are going to connect later.
>
> To increase readability in Lemma 1 (which is now Lemma 3), we have added a more substantial natural language explanation to the beginning of the proof to explain why the coarser partitions can be expressed as unions of sets at the finest partition level — i.e more written explanation for what is now equation 20. We have also added more written explanation before what is now equation 22 (used to be equation 20), where we explain how the disjointness of the $A_{\pmb{j},b_n}$ sets is used. Additionally, before the equation immediately following equation 22, we have added an explanation of how we use that the sets in  $\mathcal{S}_k$ partition $[0,1)^{d}$. We believe making these additional explanations in natural language will help enhance readability.
>
> In the new Lemma 4, we have also worked to enhance readability by following your guidance on assigning complicated expressions that do not change for large sections of the proof to temporary symbols, and including supplementary natural language explanations of proof steps.
>
> **Addressing Non-Essential Changes**
>
> 1. Yes this was a typo. Thank you for pointing this out. In this paper, expectation is with respect to capital $P_0$, because capital $P_0$ is the notation we use for the data generating distribution. We have changed any expectations that were written with an underscore little $p_0$ to capital $P_0$.
> 2. We have scanned the paper for proper noun (theorem, section, equation, definition, lemma, appendix, assumption) references and capitalized them.
> 3. We have added an explanation of the $a_n \lesssim b_n$ notation to section 2.1
> 4. In the revised, stronger version of Theorem 1, this calculation (not the Markov part which is no longer needed, but the comparison of the expected risk with the defined $\epsilon_n$) is implicit in the little-o statement in the paragraph preceding the new Equation 14. To explain this, we make references to both the definition of $\epsilon_n$ (equation 10) and the upper bound on the expected risk given in Lemma 3. We believe a look at equation 10 and Lemma 3 will be sufficient for the reader to see why the little-o statement is true.
> 5. In old Equation 18, which is now Equation 20, we have added a reference to the equation where $A_{j,b_n}$ is defined.
> 6. “Let $K$ be a positive integer” has been added to Lemma 6 (which is the old Lemma 3).
> 7. In the proof of what is now Lemma 3 (old Lemma 1), we have removed the terminology “dyadic models”, and instead refer specifically to the partitions defined in what is now Lemma 6 (which was the old Lemma 3). The use of the word dyadic “model” was imprecise and this has been corrected. We are referring here specifically to the *partitions* defined in Lemma 6, which are dyadic. Thank you for helping us achieve more precise and clear language here.

---

> > ### Comment · Reviewer_feWi · 2024-11-28
> > **Looks good.**
> >
> > Thanks a lot for addressing our points.
> >
> > I see now that the paper has grown (and some new results added), but overall I find the newer version easier to follow.

---

### Decision · Action_Editor_XHmE · 2024-12-19

**Recommendation:** Accept as is

**Comment:**

The authors have adjusted the manuscript to the reviewers' satisfaction during a productive and collaborative feedback period. All three reviewers are happy with the latest version of the manuscript.

**Audience:**

All reviewers agree that this is aligned with TMLR's audience, which I also corroborate.

**Claims And Evidence:**

All three reviewers are in agreement that the claims made in this paper are supported by clear and precise theoretical results.

---

> ### Author Response · Authors · 2025-01-05
> **Camera Ready Version Submitted**
>
> We thank the Action Editor for yielding a final decision on the paper. Note that we have just submitted the camera ready version.